# NetBurst: Event-Centric Forecasting of Bursty, Intermittent Time Series

## Abstract

Forecasting on widely used benchmark time series data (e.g., ETT, Electricity, Taxi, and Exchange Rate, etc.) has favored smooth, seasonal series, but *network telemetry time series*—traffic measurements at service, IP, or subnet granularity—are instead *highly bursty and intermittent*, with heavy-tailed bursts and highly variable inactive periods. These properties place the latter in the statistical regimes made famous and popularized more than 20 years ago by B. Mandelbrot. Yet forecasting such time series with modern-day AI architectures remains underexplored. We introduce NetBurst, an event-centric framework that reformulates forecasting as predicting *when* bursts occur and *how large* they are, using quantile-based codebooks and dual autoregressors. Across large-scale sets of production network telemetry time series and compared to strong baselines, such as Chronos, NetBurst reduces Mean Average Scaled Error (MASE) by 13–605× on service-level time series while preserving burstiness and producing embeddings that cluster 5× more cleanly than Chronos. In effect, our work highlights the benefits that modern AI can reap from leveraging Mandelbrot's pioneering studies for forecasting in bursty, intermittent, and heavy-tailed regimes, where its operational value for high-stakes decision making is of paramount interest.

## 1 Introduction

Time–series forecasting (Lim & Zohren, 2021) underpins critical decisions across domains, from finance and economics to climate science and healthcare. Recent progress has been propelled by benchmark datasets such as ETT (Zhou et al., 2021), Electricity (Trindade, 2015), Taxi (Misar, 2022), and Exchange Rate (Federal Reserve, 2022). These datasets share a common structure: they are smooth and exhibit moderate fluctuations around a seasonally varying average, with tightly bounded variance and pronounced periodicities. Forecasters trained on such data implicitly learn the "language" of continuity and regularity—and thrive within it.

**Requirements for network telemetry.** Network telemetry time series speak a very different language. Derived from measurements of traffic and devices (e.g., routers, servers, or clients), they track demand and performance at multiple granularities: *service* (applications and protocols), *IP* (individual hosts), and *subnet* (aggregated groups of hosts). Each granularity matters operationally—service forecasts guide quality assessment, IP forecasts power anomaly detection, and subnet forecasts support traffic engineering. Yet across all levels, telemetry is *highly bursty and intermittent*: burst sizes, durations, and lull periods all follow heavy-tailed laws. As emphasized in (Wierman, 2023; Willinger et al., 2005), "heavy tails are more normal than the Normal"—extreme events are not rare exceptions but defining features of these distributions. This places network telemetry in the statistical regimes that were originally shown to be of practical relevance by Mandelbrot and include fractals, self-similar scaling, $1/f$ noise, and long-range dependence—phenomena observed not only in networking but also in finance, weather, and economics datasets.

**Limitations of existing forecasters.** Today's state-of-the-art models collapse under these requirements. Transformer-based forecasters such as Chronos (Ansari et al., 2024) and Lag-Llama (Rasul et al., 2023a) conflate irregular burst timing with extreme magnitudes, smoothing away the very events operators care about. Continuous-time point-process models capture sparsity but fail to handle heavy-tailed magnitudes and long-range dependence. Even advanced tokenizers, like Chronos's uniform binning, waste resolution on dense mid-ranges while erasing fidelity in the tails. The result

is forecasting errors up to three orders of magnitude larger on production telemetry datasets such as PINOT (Beltiukov et al., 2023) and MAWI (Maw, 2025) (Table 1).

**Our proposal.** This gap motivates a new paradigm. While the statistical properties of heavy-tailed, intermittent time series are well documented (Nair et al., 2013; Resnick, 2007; Beran, 1994), how to forecast them with modern architectures remains scientifically underexplored (Hasan et al., 2023). In essence, by presenting NETBURST, an event-centric framework that reframes forecasting as disentangling *when* bursts occur and *how large* they are, our work is an instance of *Mandelbrot meets AI*. By combining distribution-aware quantile tokenization with dual autoregressive models, NETBURST allocates capacity to rare, high-impact events while preserving scalability. In doing so, we adapt foundation-style forecasters from the smooth, seasonal regimes of past benchmarks to the bursty, intermittent, heavy-tailed domains where operational impact lies.

**Contributions.**

- We introduce NETBURST, an event-centric forecasting framework that separates inter-burst gaps (timing) from burst intensities (magnitude), tokenized via quantile-based codebooks and modeled with dual autoregressors.

- We demonstrate substantial improvements over state-of-the-art forecasters on large-scale network telemetry datasets (PINOT, MAWI) across service, IP, and subnet granularities. On service-level traces, NETBURST reduces MASE to 0.0766 (PINOT) and 0.0762 (MAWI), yielding improvements ranging from 13–605× compared to existing SOTA forecasters.

- We show that NETBURST preserves burstiness: large pointwise error reductions are achieved without distorting distributional fidelity. Across subnet-level traces, NETBURST improves Wasserstein distance (WD) (Panaretos & Zemel, 2019) by roughly 2–3× relative to strong sequence forecasters, while on service-level traces its WD remains competitive with the best baselines.

- We evaluate embedding quality, showing that NETBURST improves clustering quality: silhouette scores (Shahapure & Nicholas, 2020) improve by more than 5× at small $k$ compared to baselines.

## 2 BACKGROUND AND MOTIVATION

**Network telemetry data is highly bursty and intermittent.** Recent progress in time–series forecasting has been driven by benchmarks such as ETT (Zhou et al., 2021), Electricity (Trindade, 2015), Taxi (Misar, 2022), and Exchange Rate (Federal Reserve, 2022). These datasets capture discrete-time temporal fluctuations around a mean and seasonal dynamics (Box et al., 1978), and models that perform well on them often rely on implicit assumptions of smooth and cyclic or periodic behavior. For example, the ETT dataset records electricity transformer temperatures and loads at fixed intervals, exhibiting clear daily and weekly cycles. Similarly, the New York City Taxi dataset tracks the number of rides over time, showing pronounced diurnal and weekly patterns. These benchmarks are characterized by bounded variance, non-bursty activity, and pronounced periodic patterns.

However, these properties fail to capture the statistical features of real-world *network telemetry time series*—measurements collected from production network traffic or devices (e.g., routers, end-hosts) to monitor performance, reliability, and security. Network telemetry metrics (e.g., packet or byte counts) are often aggregated at different spatial granularities—*service* (applications and protocols), *IP* (individual hosts), and *subnet* (groups of hosts)—to provide a holistic view of network state. Forecasting at each granularity serves distinct operational goals: service-level forecasts inform quality assessment and classification, IP-level forecasts support anomaly detection and intrusion monitoring, and subnet-level forecasts guide traffic engineering and capacity planning Guok et al. (2025).

In contrast to existing benchmarks, network telemetry time series are *highly bursty and intermittent*, with burst sizes, burst durations, and lull periods all typically following heavy-tailed distributions (Nair et al., 2013). Most activity is modest, but rare events can be orders of magnitude larger than the mean and dominate both the magnitude and variability of the traffic. Figure 1 quantifies these statistical differences[1]. Figure 1a shows the CCDF of Fano factors (variance-to-mean ratios) Fano (1947)

---

[1]We observe similar trends for other benchmarks and granularities for temeletry data

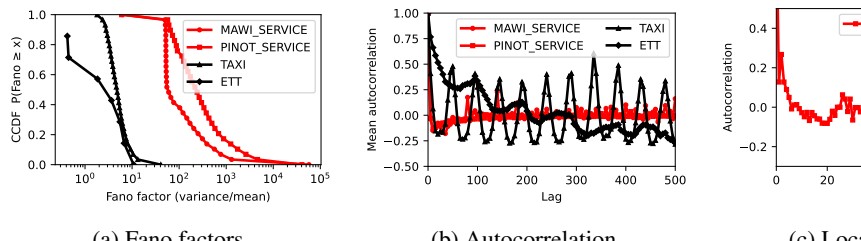

(a) Fano factors        (b) Autocorrelation        (c) Local example.

Figure 1: Current forecasting benchmarks (ETT and Taxi) vs. network telemetry data (more bursty service time series): (a) shows benchmarks confined to narrow ranges, while network telemetry exhibits extreme variability. (b) reveals strong periodicity in benchmarks but flat structure in telemetry. (c) shows only localized, phase-misaligned micro-patterns that cancel when aggregated.

across datasets: while benchmark series concentrate within narrow bounds, telemetry time series exhibit heavy-tailed behavior, reflecting entities whose traffic is dominated by rare, extreme bursts. Figure 1b depicts empirical autocorrelation functions Box et al. (2015) across datasets. While benchmarks exhibit clear periodic patterns indicative of seasonal cycles, telemetry traces show no signs of periodic behavior, and the autocorrelations remain flat at scale. For example, when zooming-in on the intial portion of the empirical autocorrelation function for the telemetry time series Pinot-Service, Figure 1c shows localized micro-patterns, which are phase-misaligned and cancel when aggregated.

Table 1 highlights the collapse of state-of-the-art forecasters when applied to bursty, intermittent telemetry. Models tuned to smooth, seasonal datasets fail by one to three orders of magnitude on PINOT and MAWI precisely because they smooth over rare bursts or default to trivial predictions. DeepAR (Salinas et al., 2020) minimizes loss in sparse regimes by predicting all zeros, while N-BEATS (Oreshkin et al., 2020), designed around periodic templates, extracts no meaningful structure. Even models like Chronos (Ansari et al., 2024) and Lag-Llama (Rasul et al., 2023a), which claim to succeed without time-series–specific biases, implicitly assume continuity and bounded variance: Chronos's "agnostic to time" design works on smooth benchmarks but erases timing and tail fidelity in telemetry, and its reported zero-shot generalization holds only for non-bursty regimes. The implication is clear: forecasting telemetry requires elevating bursts to first-class modeling units, rather than treating the series as continuous fluctuations around a seasonal mean.

| Dataset | Chronos | DeepAR | Lag-Llama | N-BEATS |
|---|---|---|---|---|
| Electricity (15m) | 0.2263 | 0.4080 | 0.2036 | 0.4243 |
| Taxi (30m) | 0.5964 | 0.7258 | 0.6213 | 0.5851 |
| ETT-M2 (hourly) | 0.1025 | 0.1213 | 0.150 | 1.0034 |
| Weather (daily) | 0.3564 | 0.5759 | 0.4154 | 1.0455 |
| Exchange Rate (daily) | 0.0253 | 0.0567 | 0.0432 | 0.0581 |
| Pinot (Service, 100ms) | 18.1282 | 1.0019 | 0.9197 | 46.35 |
| Pinot (IP, 1s) | 782.3511 | 2.2039 | 2.0043 | 86.47 |
| Pinot (Subnet, 1s) | 1640.4 | 1.9807 | 1.7953 | 95.53 |
| MAWI (Service, 100ms) | 2.1782 | 0.9994 | 1.0161 | 35.5556 |
| MAWI (IP, 1s) | 1629.3 | 8.8251 | 34.5436 | 44.9187 |
| MAWI (Subnet, 1s) | 13.4595 | 5.1334 | 13.7958 | 98.6089 |

Table 1: SOTA performance (MASE) on existing benchmarks vs. network telemetry.

**Sparsity and entanglement break sequence models.** Sparse time series exacerbate this challenge. Bursts are rare and widely separated, so capturing them requires long context windows. However, naïvely extending sequence length is wasteful: most tokens correspond to idle or noisy intervals. Models either dilute bursts among zeros or learn shortcuts, such as predicting zero everywhere. This explains the misleadingly low errors of DeepAR on some sparse series—it minimizes loss by ignoring rare events entirely. Even when bursts are predicted, monolithic models conflate *when* they occur and *how large* they are. These two distributions—inter-burst gap (IBG) and burst intensity (BI)—have distinct statistical behavior. Sequence models that treat them jointly allocate capacity inefficiently, leading to unstable forecasts. As our later Oracle analyses confirm, IBG dominates error on sparse Service traces, while BI dominates on denser IP/Subnet traces. This motivates disentangling timing from magnitude, using separate prediction streams with light coupling.

**Uniform binning under-represents tails.** Tokenized foundation models such as Chronos (Ansari et al., 2024) introduce discretization, but use uniform bins. This allocates resolution evenly across the value range, oversampling dense low-magnitude fluctuations and undersampling rare, high-magnitude bursts. On existing datasets, this is adequate, but in telemetry, it systematically erases the

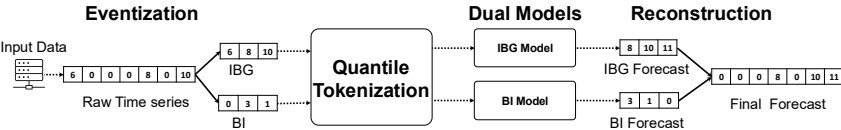

Figure 2: NETBURST pipeline. Raw telemetry series are eventized into inter-burst gaps (IBG) and burst intensities (BI), discretized with quantile-based tokenizers, and modeled with separate autoregressive forecasters. The two streams are then recombined during reconstruction to produce byte-count forecasts that preserve sparsity and burst fidelity.

very events that dominate tail error. Table 1 illustrates this directly: despite pretraining, Chronos collapses on network telemetry data. These results call for distribution-aware codebooks that allocate bins according to probability mass, preserving tail fidelity.

**Embeddings collapse under baselines.** Forecasting performance is only part of the story. Operators often need to cluster entities, detect anomalies, or transfer models across contexts. Embeddings from existing models collapse in these regimes: they show high anisotropy and poor clustering quality, dominated by trivial small fluctuations rather than bursts. This undermines downstream tasks as much as forecasting. We later show that event-centric embeddings—derived from IBG and BI token streams—yield richer and more isotropic representations.

**Beyond networking telemetry.** This bursty, intermittent behavior is not unique to networking; it reflects statistical phenomena extensively studied by Mandelbrot, who introduced concepts such as *fractals*, *self-similar scaling*, $1/f$ *noise*, and *long-range dependence* to describe heavy-tailed time series in domains like telecommunications, finance, economics, and weather (Kirkby, 1983; Taqqu et al., 1997; Leland et al., 2002; Verma et al., 2025). Existing state-of-the-art forecasters are likely to encounter similar challenges in these settings. Although the statistical properties of such processes are well established, how to forecast them effectively with modern architectures such as transformers remains scientifically underexplored despite their operational importance.

## 3 EVENT-CENTRIC FORECASTING WITH NETBURST

**Overview: From sequences to events.** Forecasting network telemetry time series with conventional sequence models fails for several reasons established in §2: sparsity caused by bursty and intermittent behavior makes long contexts inefficient, timing- and magnitude-related aspects are entangled in ways that destabilize predictions, uniform binning ignores or discounts heavy-tailed extremes, and learned embeddings collapse under skewed activity.

To address these issues, we adopt a different guiding principle: reframe forecasting as an *event prediction problem* rather than a direct sequence-value regression task. Instead of predicting raw time series values (i.e., byte counts) at every step, we represent telemetry time series as a sequence of bursts, each defined by **when** it occurs and **how large** it is. This decomposition parallels the structure of a marked Hawkes process Hawkes (1971): inter-burst gaps (IBG) capture event timings, while burst intensities (BI) serve as continuous marks. Unlike traditional Hawkes models (Zuo et al., 2020; Meng et al., 2024), however, our formulation remains scalable, token-based, and able to accommodate heavy-tailed magnitudes.

Concretely, our pipeline proceeds in four steps as shown in Fig 2. (i) *Eventization*: convert time series into IBG and BI streams, compressing idle periods while preserving burst statistics. (ii) *Quantile tokenization*: discretize each stream with distribution-aware codebooks that allocate resolution to rare, extreme bursts. (iii) *Dual autoregressive models*: predict IBG and BI independently with autoregressive transformers, disentangling timing from magnitude while preserving local dependence. (iv) *Reconstruction*: recombine predicted events into byte-rate forecasts that retain both timing and magnitude fidelity. This event-centric design operationalizes the statistical properties documented in §2 into a forecasting framework that directly addresses the failure modes of existing models while preserving the self-exciting burst dynamics inherent to network telemetry data.

**Eventization: Separating timing from magnitude.** Directly modeling network telemetry time series is problematic: most windows contain little or no traffic, so long contexts waste capacity on zeros; and when bursts do occur, their *timing* and *magnitude* are entangled in ways that destabilize forecasts (§2). To address this, we reformulate forecasting as an event sequence prediction task.

Concretely, given a series $s_{1:T}$, we fix an activity threshold $T_{\text{act}}$ and declare a *burst* whenever consecutive windows exceed this threshold. Each burst $[\tau_k, \rho_k]$ is summarized by two values: the *inter-burst gap* $\text{IBG}_k = \tau_k - \tau_{k-1}$ (with $\text{IBG}_1 = \tau_1$) capturing **when** the burst occurs, and the *burst intensity* $\text{BI}_k = \sum_{t=\tau_k}^{\rho_k} s_t$ capturing **how large** the burst is.

This transformation collapses long idle stretches into a single large IBG and replaces each burst with a compact BI mark. The resulting eventized sequence is far shorter and less noisy than the raw time series, yet it retains exactly the information needed for forecasting: the distribution of gaps between bursts and the heavy-tailed distribution of burst sizes. By disentangling timing from magnitude, eventization not only reduces wasted context but also makes the self-exciting burst structure of network traffic explicit.

**Quantile Tokenization: Preserving Heavy Tails.** Uniform binning, as used in Chronos (Ansari et al., 2024), wastes resolution on dense mid-range values and severely under-represents the rare, extreme events that dominate heavy-tailed telemetry time series. This mismatch leads baselines to smooth away precisely the bursts that matter most in practice (§2).

To address this issue, NETBURST replaces uniform binning with *quantile-based codebooks*. Instead of dividing the value range evenly, we discretize each stream so that every token corresponds to an equal fraction of probability mass. Specifically, we construct global quantile codebooks $Q^{\text{IBG}}, Q^{\text{BI}}$ such that each bin holds approximately equal mass on the training data.

At training time, each observed inter-burst gap (IBG) or burst intensity (BI) is mapped to a token index $z_k^{\text{IBG}}, z_k^{\text{BI}} \in \{1, \ldots, B\}$ according to its quantile bin. Forecasting then reduces to next-token prediction, and reconstruction replaces each token with its bin centroid $\hat{Q}^{\text{IBG}}, \hat{Q}^{\text{BI}}$. This way, we preserve the shape of heavy tails, stabilize learning, and enable NETBURST to represent rare but operationally critical events as faithfully as routine fluctuations.

**Dual Models: Forecasting timing and magnitude separately.** Even after eventization and quantile tokenization, forecasting remains unstable if timing and magnitude are forced through a single prediction head. Models such as DeepAR or Lag-Llama conflate these dimensions, causing errors in one component to cascade into the other.

To address this problem, NETBURST employs two autoregressive transformer models: one for inter-burst gaps (IBG) and one for burst intensities (BI). Each model predicts its quantile-tokenized stream independently, i.e., $f_\theta : (z_{<k}^{\text{IBG}}) \mapsto p_\theta(z_k^{\text{IBG}} \mid z_{<k}^{\text{IBG}})$ and $g_\psi : (z_{<k}^{\text{BI}}) \mapsto p_\psi(z_k^{\text{BI}} \mid z_{<k}^{\text{BI}})$.

*Training.* Each model minimizes next-token cross-entropy with teacher forcing: $\mathcal{L}_{\text{IBG}} = \sum_k -\log p_\theta(z_k^{\text{IBG}} \mid z_{<k}^{\text{IBG}})$ and $\mathcal{L}_{\text{BI}} = \sum_k -\log p_\psi(z_k^{\text{BI}} \mid z_{<k}^{\text{BI}})$.

*Inference.* During decoding, IBG and BI tokens are generated autoregressively, $\hat{z}_k^{\text{IBG}} \sim p_\theta(\cdot \mid \hat{z}_{<k}^{\text{IBG}})$ and $\hat{z}_k^{\text{BI}} \sim p_\psi(\cdot \mid \hat{z}_{<k}^{\text{BI}})$, then mapped back to numeric values through their quantile codebooks $Q^{\text{IBG}}$ and $Q^{\text{BI}}$. This separation stabilizes learning, avoids error entanglement, and allocates modeling capacity where it is most needed.

**Reconstruction: From events back to series.** The final step is to transform predicted event streams into a standard byte-count series usable by networking tasks. Predicted IBGs are accumulated into absolute event times, $\hat{\tau}_k = \hat{\tau}_{k-1} + \hat{\text{IBG}}_k$, and paired with their predicted burst intensities, $\hat{\text{BI}}_k$. By default, we allocate each predicted burst entirely to its starting window ("spike placement"), though richer intra-burst kernels can be substituted. This reconstruction closes the loop: idle periods collapse into long IBGs, bursts are preserved with their predicted magnitudes, and the output recovers a time series that retains the statistical properties of the original telemetry time series and can be used by downstream task that require forecasting.

# 4 EVALUATION

## 4.1 EXPERIMENTAL SETUP

**Datasets and preprocessing.** We evaluate NETBURST on both standard time-series forecasting benchmarks and large-scale networking telemetry datasets. The benchmarks include Electricity, Taxi, ETT, Exchange Rate, and Weather, each preprocessed at their originally reported sampling

resolutions. For network telemetry, we use two major sources: PINOT and MAWI. Specifically, the PINOT dataset contains packet traces (1.98 billion packets) collected from a campus gateway router for the week of Dec 10th 2023 over 13 15-minute intervals, while the MAWI dataset contains packet traces (23 million packets) collected from an ISP backbone router on 24th May 2025, over a single 15-minutes interval. From these traces, we generate time series representing the number of bytes transferred, aggregating packets at different temporal granularities (from milliseconds to minutes) and spatial levels (service, IP, and /24 subnet). Unless otherwise specified, we report results at 100 ms resolution for service time series and 1 s IP and subnet time series. These intervals are both operationally relevant (service level classification and quality estimation tasks require faster decision-making, and vice versa) and, as shown in Figure 7 (Appendix), strike a balance in capturing traffic dynamics: smaller windows expose fine-grained bursts but increase sparsity, whereas larger windows suppress variability and obscure burst structure. Table 2 (Appendix) summarizes the number of time series at Service, IP, and Subnet granularities at these two temporal resolutions.

**Baselines.** We compare NETBURST against both foundation-style and classical forecasters to ensure coverage of the most competitive and representative approaches. Chronos-T5 (Ansari et al., 2024) is included as a strong foundation baseline: it uses uniform binning and large-scale pretraining, making it the most direct point of comparison for our quantile-based tokenization. Lag-Llama (Rasul et al., 2023b) represents recent autoregressive foundation models that emphasize cross-dataset transfer, allowing us to test whether broad pretraining and simple AR decoding suffice for bursty, event-driven telemetry data. DeepAR (Salinas et al., 2020) provides a probabilistic RNN baseline with likelihood-based training, probing whether explicit probabilistic modeling captures tails and uncertainty relevant to rare-event forecasting. Finally, N-BEATS (Oreshkin et al., 2020) serves as a strong non-attention baseline with trend/seasonality inductive biases, testing whether generic deep residual architectures perform adequately on data with weak global seasonality but heavy-tailed local burst structure. All baselines are tuned with comparable hyperparameter budgets to ensure fairness in evaluation (see **Appendix A** for details).

**Training protocols and implementation.** To ensure fairness, all models follow identical protocols. Time series are split chronologically into 70% training/context, 20% testing, and 10% validation. Eventization is applied post-split: IBG and BI sequences are recomputed for each $T_{act}$, with quantile codebooks fit only on training data to prevent leakage. At inference, IBG and BI heads decode autoregressively, and predictions are mapped via codebook centroids. Metrics are averaged across entities and reported by granularity and threshold. We use Adam (Kingma & Ba, 2014) with learning rate $1 \times 10^{-4}$, batch size 32, and early stopping on validation loss. NETBURST uses a 12-layer seq2seq transformer (hidden size 512), following Chronos but replacing uniform bins with 4096-bin quantile codebooks to preserve both small fluctuations and large bursts. All baselines use the same optimizer, batch size, and early stopping policy. Additional hyperparameters are summarized in Table 5 in the Appendix.

**Evaluation metrics.** We compute Absolute Scaled Error (ASE) to capture pointwise accuracy relative to ground truth. We report both the mean of ASE (MASE) and its full distribution across entities and horizons, which reveals where errors concentrate and highlights rare but operationally critical failures. To mitigate sparsity bias, we compute MASE only on events, following prior work (Zuo et al., 2020; Meng et al., 2024). Finally, we use 1-Wasserstein Distance (WD) to assess distributional fidelity by jointly evaluating burst timing and magnitude.

## 4.2 NETBURST VS. BASELINES

**Does NETBURST offer significant gains on bursty and intermittent network telemetry data?**

We begin with the most challenging regime: service-level time series at 100 ms resolution. Figure 3 highlights forecasting performance for both PINOT and MAWI. On PINOT service data, NETBURST achieves a MASE of 0.0766, while the strongest baseline (DeepAR) records 1.00—a 13× higher MASE. Lag-Llama fares worse at 2.29 (30× higher), and Chronos-T5 and N-BEATS collapse catastrophically at 18.1 (237×) and 46.3 (605×) respectively. A similar pattern holds for MAWI service traces, where NETBURST reaches 0.0762 versus

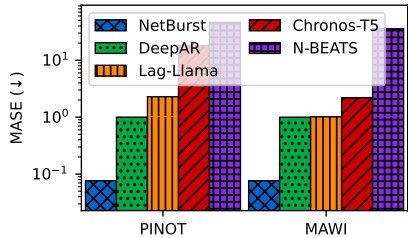

Figure 3: NETBURST vs. Baselines

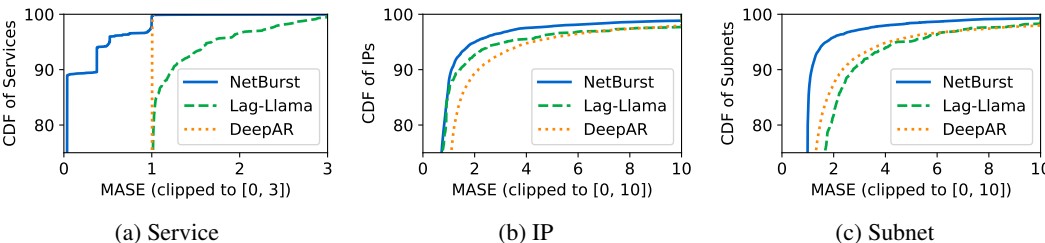

(a) Service           (b) IP           (c) Subnet

Figure 4: CDF of MASE losses shows NETBURST has fewer examples where losses were large, demonstrating effectiveness in predicting rare events.

DeepAR's 1.00 (13× higher) and Lag-Llama's 1.02 (13.3× higher), with Chronos-T5 at 2.18 (28.6×) and N-BEATS at 35.6 (467×). These results confirm our central hypothesis: NETBURST delivers order-of-magnitude gains in the most bursty and intermittent settings, where prior SOTA models fail.

Table 3 (Appendix) generalizes this comparison across all granularities. On PINOT IP-level data, NETBURST reduces error to 0.90, compared to 2.20 (2.5× higher) for DeepAR and 2.62 (2.9× higher) for Lag-Llama. On PINOT subnet data, NETBURST achieves 0.96, again halving the error relative to DeepAR and Lag-Llama (both at ∼2.0). For MAWI, NETBURST records 7.15 MASE at the IP level versus 8.83 (1.2× higher) for DeepAR and 34.5 (4.8×) for Lag-Llama. At the subnet level, NETBURST attains 4.35 compared to 5.13 (1.2×) for DeepAR and 13.8 (3.2×) for Lag-Llama. While the margins are smaller at coarser granularities—reflecting the reduced sparsity of aggregated flows—NETBURST consistently outperforms baselines. Importantly, models like Chronos-T5 and N-BEATS collapse in nearly every case, producing errors that are two to three orders of magnitude larger than NETBURST.

An examination of the error distributions further illustrates this gap. Figure 4 plots the CDF of absolute scaled errors for PINOT across all granularities (linear scale, excluding the worst baselines Chronos and N-BEATS). NETBURST shifts the error mass consistently leftward compared to both DeepAR and Lag-Llama, yielding tighter and lower error distributions. A notable pathology emerges: DeepAR collapses in sparse regimes by learning a degenerate "always zero" forecast. While this forecast reduces error superficially on quiescent intervals, it fails catastrophically on bursts, exposing a vulnerability of sequence models to shortcut learning in sparse domains. NETBURST, by contrast, maintains robustness across both sparse and dense regions.

**Does NETBURST preserve burstiness while decomposing forecasts?** A critical concern is whether decomposing traffic into inter-burst gaps and burst intensities compromises burstiness in the reconstructed series. We address this by reporting the Wasserstein distance (WD), which evaluates distributional fidelity (See Table 4 in Appendix). On the most challenging service-level time series, NETBURST achieves WD values of 0.0006 (PINOT) and 0.0009 (MAWI), which are comparable to DeepAR (0.0006 / 0.0001) and Lag-Llama (0.0005 / 0.0002). This demonstrates that the large MASE gains of NETBURST on bursty intermittent data do not come at the expense of distorted distributional structure. At coarser granularities, NETBURST in fact improves WD relative to strong baselines: on PINOT subnet data, it attains 0.0067, outperforming Lag-Llama (0.0101) and DeepAR (0.0105), and on MAWI subnet, it reaches 0.0112, improving on Lag-Llama (0.0332) and DeepAR (0.0254). Taken together, these results show that NETBURST's event-centric decomposition preserves or enhances burstiness fidelity across all granularities, while delivering substantial pointwise error reductions where they matter most.

### 4.3 ABLATION STUDIES

**Do quantile codebooks improve fidelity over uniform binning?** A central design choice in NETBURST is to replace uniform binning, as employed by Chronos, with distribution-aware quantile-based tokenization. Quantile codebooks allocate resolution according to probability mass, ensuring that rare, high-magnitude bursts are well represented, while uniform binning wastes tokens on dense regions and undersamples the tails. To isolate this effect, we evaluate three models at each granularity (Service, IP, Subnet): (i) Chronos as-is (i.e., using uniform binning), (ii) a variant of Chronos

that replaces uniform bins with quantile-based codebooks while keeping the rest of the architecture unchanged, and (iii) NETBURST, which combines quantile-based tokenization with stream decomposition via dual models.

Figures 5a show that moving from uniform to quantile binning consistently lowers MASE and WD (not shown for brevity), with the largest gains on Service traces where burstiness is most severe. Notably, the performance gap between models (ii) and (iii) diminishes at coarser granularities, indicating that quantile-based tokenization contributes the bulk of the improvement for IP and Subnet. In contrast, at the finer and more bursty Service level, the full combination of quantile binning and stream decomposition is most effective. These results confirm that quantile-based tokenization is not an incidental detail but a critical design choice, while disentangling streams adds further leverage in the most challenging sparse regimes.

**Where does residual error come from: timing or magnitude?** Finally, we ask whether residual error is primarily due to imperfect modeling of inter-burst timing or burst magnitudes. To answer this, we evaluate three variants: (i) the trained NETBURST model, (ii) Oracle-IBG, which substitutes ground-truth inter-burst gaps while keeping predicted burst intensities, and (iii) Oracle-BI, which substitutes ground-truth burst intensities while keeping predicted inter-burst gaps. Figure 5b shows that on IP and Subnet data, Oracle-BI reduces WD more than Oracle-IBG, implicating magnitude alignment as the dominant residual source. On Service traces, however, Oracle-IBG closes more of the gap, indicating that timing dominates in the sparsest regime. These results suggest that future improvements should target stronger temporal modeling for Service, while magnitude modeling is the higher-leverage direction for coarser granularities.

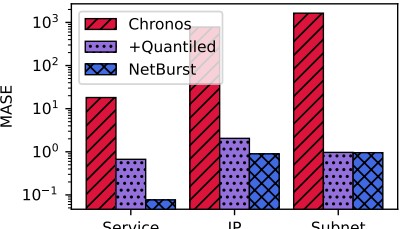

(a) Quantile codebook ablation.

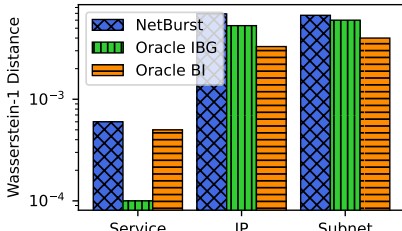

(b) Comparison with oracles.

Figure 5: Ablation study.

### 4.4 REPRESENTATIONAL QUALITY OF EMBEDDINGS

Beyond forecasting accuracy, it is critical to assess whether NETBURST's event-centric embeddings provide richer representational quality than baseline models. To this end, we use a mix of volumetric traces from the PINOT dataset. These traces serve as "cross-traffic profiles" (CTPs) that enable emulating realistic network dynamics in controlled settings (Daneshamooz et al., 2025). Clustering CTPs with similar causal effects on network applications competing for limited resources at a bottleneck link is essential for scaling meaningful data generation in such emulators. We therefore evaluate the quality of NETBURST embeddings by studying their ability to group 50k CTPs subsampled from PINOT's campus network traces.

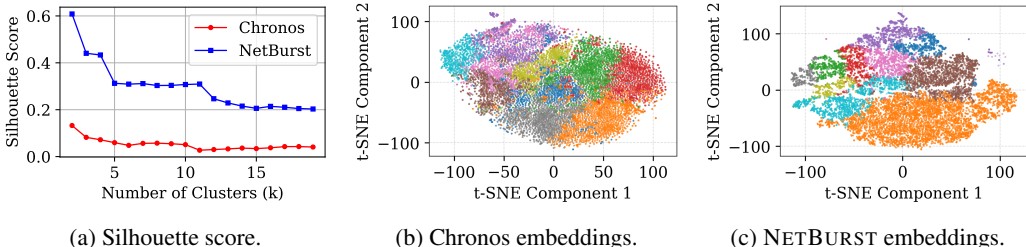

(a) Silhouette score.     (b) Chronos embeddings.     (c) NETBURST embeddings.

Figure 6: Representational quality of embeddings extracted from IP-level traces. NETBURST reduces anisotropy and improves clustering, yielding semantically richer embeddings.

We apply $k$-means clustering on the embeddings and measure the silhouette score as $k$ varies. Higher silhouette scores indicate clearer grouping of similar time series, while lower scores suggest overlapping or poorly separated clusters. Figure 6(a) shows that Chronos embeddings yield consistently

low silhouette scores across all $k$, whereas NETBURST embeddings achieve substantially higher values, improving by more than $5\times$ at small $k$. To qualitatively assess this effect, we project embeddings into two dimensions using t-SNE. Figures 6(b) and 6(c) illustrate the clustering behavior of Chronos and NETBURST. Chronos embeddings collapse into overlapping regions with weak separation, consistent with their higher anisotropy and low silhouette scores. In contrast, NETBURST embeddings form distinct, well-structured groups that align with different CTP behaviors, reflecting stronger clustering capability. Taken together, these results show that NETBURST not only improves forecasting accuracy but also produces semantically richer embeddings that transfer more effectively to downstream grouping and classification tasks.

## 5 DISCUSSION AND BROADER IMPACT

**Model transferability.** Our transferability analysis (see § A.1) builds on the hypothesis that the bursty, self-similar nature of network telemetry data allows statistical structure to repeat across granularities. Specifically, once small fluctuations are suppressed, time series at coarser levels (IP, subnet) begin to resemble their fine-grained counterparts (service), enabling cross-granularity reuse. Empirically, we find that service-pretrained models can be adapted to IP and subnet series by thresholding low-activity windows: distributional similarity improves monotonically with higher thresholds, and the transferred model nearly matches the performance of specialized per-level models on both MASE and WD. These results provide the first concrete evidence that a single event-centric model, trained at fine granularity, can generalize across spatial scales with minimal fine-tuning. Beyond cost savings, this supports the broader vision of NETBURST as a foundation model for telemetry—one capable of leveraging fractal-like scaling to unify forecasting across heterogeneous granularities and operational settings. Realizing this promise more fully, especially for other self-similar time series beyond networking, remains an exciting direction for future work.

**Operational practice.** Accurate burst forecasting directly supports operators in congestion anticipation, anomaly detection, and capacity planning. Event-centric embeddings also provide transferable representations for downstream tasks such as traffic classification and policy-driven control. More broadly, viewing forecasting as representation learning creates opportunities for transfer across datasets and domains.

**Beyond networking.** The statistical regime we target—bursty, intermittent, heavy-tailed time series—extends well beyond networking. Financial transactions, reliability logs, epidemic outbreaks, and climate extremes all exhibit irregular bursts separated by long idle periods. By decoupling timing from magnitude, NETBURST offers a general event-centric paradigm for forecasting processes where rare, high-impact events matter more than averages.

**Limitations & future work.** NETBURST currently models bursts as spikes with aggregated intensity, ignoring intra-burst structure; richer kernels or learned profiles could capture within-burst dynamics. Eventization relies on fixed thresholds, whereas adaptive or learned thresholds may yield more flexible decompositions. Finally, residual errors are dominated by timing (Oracle-IBG), motivating exploration of hierarchical Hawkes-like models or uncertainty-aware predictors.

## 6 CONCLUSION

On large-scale telemetry datasets (PINOT, MAWI), NETBURST reduces MASE by $13\text{--}605\times$ compared to state-of-the-art forecasters at the bursty and intermittent service-level granularity, while consistently preserving burstiness across all levels, achieving up to $3\times$ better Wasserstein distance at the subnet level. These gains, concentrated on rare, high-intensity bursts, stem from NETBURST's event-centric design: reframing forecasting as predicting *when* bursts occur and *how large* they are, using quantile-based codebooks and dual autoregressors. By allocating capacity to extreme events while retaining distributional fidelity, NETBURST overcomes the limitations of transformer forecasters and point-process models that smooth away or collapse under heavy tails. This work establishes forecasting of bursty, intermittent, heavy-tailed time series—long recognized in Mandelbrot's work on self-similar scaling phenomena—as both an operationally critical and scientifically under-explored challenge, and positions NETBURST as a step toward foundation models that learn the language of rare extremes across networking, finance, climate, and beyond.

## REPRODUCIBILITY STATEMENT

We have made every effort to ensure that our work is reproducible. An anonymized repository accompanies this submission and includes: (i) pointers to the raw, publicly accessible MAWI dataset used in our experiments (noting that raw PINOT data is not publicly available); (ii) pointers to fully anonymized, pre-processed datasets used for analysis; and (iii) pre-processing scripts, model training code, and experiment configurations corresponding to the evaluations reported in § 4. The anonymized repository for NetBurst can be found at `https://anonymous.4open.science/r/NetBurst-F135/`. Together with the methodological details provided in the main text and additional clarifications in the appendix, these resources are intended to facilitate faithful reproduction of our results.

## ETHICS STATEMENT

All authors have read and adhered to the ICLR Code of Ethics. This research does not involve human subjects directly. However, packet traces may contain personally identifiable information (PII), which raises privacy considerations. We have therefore taken extensive measures to ensure responsible data handling and compliance with institutional and ethical standards.

Our data collection and research protocols were reviewed and exempted by the university's Institutional Review Board (IRB). We have provided the exemption letter to the program committee. Because the dataset originates from an operational campus network, the collection and research processes were also reviewed by a separate institutional committee comprising campus stakeholders and IT experts, which approved the use of this dataset for research purposes. All researchers accessing the dataset completed mandatory training in research ethics and data privacy.

**Anonymization.** To minimize privacy risks, we only collect a minimal subset of packet header fields. Packet payloads are discarded beyond RTP header extensions. Internal (campus) IP addresses are anonymized in a prefix-preserving manner at the building level using a modified ONTAS system (Kim & Gupta, 2019), implemented directly in the hardware switch via P4. This ensures that traffic is anonymized *before* reaching the collection server. External (non-campus) IP addresses are preserved to enable analysis of Autonomous Systems and application-level behavior (e.g., Zoom). While such addresses could theoretically be linked to users, they are typically associated with large ISPs or enterprises and are dynamically assigned, making re-identification infeasible. No effort was made to deanonymize or identify any individual user.

**Data management.** The campus dataset is stored in a secure infrastructure managed by professional IT staff. Access is strictly limited to IRB-approved researchers and authorized IT personnel. The dataset cannot be moved or copied outside the secure environment, ensuring compliance with institutional data protection requirements.

We believe these safeguards adequately address potential ethical concerns regarding privacy, security, and responsible research conduct.

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

# A    APPENDIX



| (a) Service-level traces | (b) IP-level traces | (c) Subnet-level traces |

Figure 7: Fano Factor of the 95th-percentile Service, IP, and Subnet series under varying thresholds and window sizes. Smaller windows increase variance and sparsity, while larger windows reduce burst detail. This motivates the choice of 100 ms and 1 s windows for evaluation.

| Dataset | Service | IP | Subnet |
|---|---|---|---|
| PINOT (100 ms) | 20.5M | 1.81M | 737K |
| PINOT (1 s) | 7.44M | 931k | 361K |
| MAWI (100 ms) | 1.35M | 753K | 157K |
| MAWI (1 s) | 1.08M | 743K | 135K |

Table 2: Networking datasets.

Table 3: Relative MASE for existing SOTA forecasters versus NETBURST (smaller is better). Each entry shows how much worse the baseline error is compared to NETBURST (first column) on the same dataset.

| Dataset | NETBURST (MASE) | DeepAR | Lag-Llama | Chronos | N-BEATS |
|---|---|---|---|---|---|
| **PINOT-Service** | 0.0766 | 13.1× | 29.8× | 236.8× | 605.1× |
| **PINOT-IP** | 0.8963 | 2.46× | 2.92× | 872.8× | 96.5× |
| **PINOT-Subnet** | 0.9553 | 2.07× | 2.07× | 1717.2× | 100.0× |
| **MAWI-Service** | 0.0762 | 13.1× | 13.3× | 28.6× | 466.6× |
| **MAWI-IP** | 7.1546 | 1.23× | 4.83× | 227.8× | 6.28× |
| **MAWI-Subnet** | 4.3533 | 1.18× | 3.17× | 3.09× | 22.7× |

## A.1    MODEL TRANSFERABILITY

**Hypothesis.** The bursty and intermittent temporal dynamics of network telemetry time series, often referred to as the fractal or self-similar nature of network traffic (Willinger et al., 2002), enables effective cross-granularity transfer: a model pre-trained at a finer granularity (e.g., service-level traces) can be adapted to coarser granularities (e.g., IP, subnet) by thresholding small values as noise, thereby aligning their statistical structure. If true, this would allow operators to reuse a single pre-trained model across granularities with minimal fine-tuning, avoiding the cost of per-granularity pre-training.

**Does thresholding reveal distributional similarity across granularities?**    Sparsity in network telemetry time series allows for a natural partitioning into "interesting" (nontrivial activity) and "uninteresting" (idle or negligible activity) windows. Crucially, what counts as interesting depends on the aggregation granularity. By thresholding activity, quantizing active windows with global quantile binning, and measuring entropy, we find that once low-activity windows are suppressed, the statistical structure self-repeats across levels. Figure 8a reports the Jensen-Shannon divergence (JSD) between service series and thresholded IP/subnet series. As the threshold $T_{act}$ increases (100,

Table 4: Wasserstein distance (WD; ↓ lower is better) on networking datasets (100 ms and 1 s).

| Dataset | NETBURST | DeepAR | Lag-Llama | N-BEATS | Chronos-T5 |
|---|---|---|---|---|---|
| **PINOT-Service** | 0.0006 | 0.0006 | 0.0005 | 0.0193 | 0.0020 |
| **PINOT-IP** | 0.0069 | 0.0080 | 0.0071 | 0.0220 | 0.4174 |
| **PINOT-Subnet** | 0.0067 | 0.0105 | 0.0101 | 0.0376 | 1.0639 |
| **MAWI-Service** | 0.0009 | 0.0001 | 0.0002 | 0.0042 | 0.0010 |
| **MAWI-IP** | 0.0253 | 0.0352 | 0.0484 | 0.0361 | 2.5553 |
| **MAWI-Subnet** | 0.0112 | 0.0254 | 0.0332 | 0.1439 | 0.0272 |

200, 300 bytes), the JSD decreases monotonically, indicating that thresholding progressively aligns coarser time series distributions with service-level distributions. This fractal behavior reaffirms the self-similarity of network traffic, extensively studied in prior work such as (Willinger et al., 2002; Leland et al., 2002; Taqqu et al., 1997), and highlights its modern implications for learning: models can, in principle, transfer across granularities.

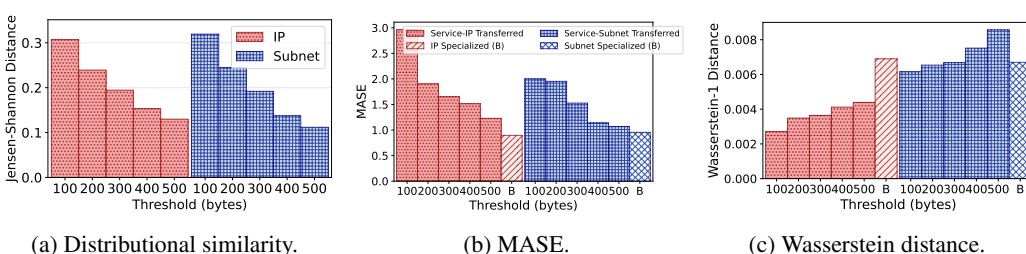

(a) Distributional similarity.  (b) MASE.  (c) Wasserstein distance.

Figure 8: MASE and WD of Service model when adapted to tokenizers based on IP and Subnet datasets under different thresholds.

**Does a service-level pretrained model transfer effectively to coarser granularities?** We next test whether this statistical similarity translates into model performance. Figure 8b reports MASE and WD for service-pretrained models transferred to IP and subnet datasets under different thresholds, compared against specialized IP/subnet-pretrained references. Two consistent trends emerge. First, as $T_{act}$ increases, MASE decreases because small fluctuations are ignored and the forecast focuses on salient bursts. Second, WD increases slightly because thresholding shifts mass into zeros and introduces residual timing or magnitude mismatches on micro-activity. Importantly, at higher thresholds (200–300 bytes), the transferred service-pretrained model nearly matches the specialized models on both MASE and WD. This demonstrates the feasibility of cross-granularity transfer: one NETBURST model pre-trained on fine-grained service data can generalize to coarser granularities with minimal fine-tuning.

**Implications.** In practice, these findings offer significant efficiency advantages. Network operators may aggregate traffic differently across settings, and retraining per-granularity models would be costly. By leveraging the fractal or self-similarity nature of network telemetry time series and applying thresholding transformations, NETBURST can be pre-trained once at fine granularity and adapted across granularities, preserving fidelity while reducing training costs. This supports the broader vision of NETBURST as a foundation model for network telemetry time series that generalizes across both temporal and spatial scales.

## A.2 LLM USAGE

Portions of this work benefited from the use of large language models (LLMs). Specifically, LLM-based code editors were employed to expedite prototyping and experimentation, and interactive chat-based interfaces were used to assist in refining and polishing the manuscript text. All substantive ideas, methodological contributions, and experimental designs originate from the authors, and the authors take full responsibility for the content and conclusions presented in this paper.

Table 5: Baseline hyperparameters (paper-aligned recommendations). Where papers specify ranges or dataset-dependent choices, we list the range/guideline rather than a single value.

| Baseline | Context length | Batch size | Hidden size | Vocabulary | Learning rate | Optimizer |
|----------|---------------|------------|-------------|------------|---------------|-----------|
| DeepAR | 512 | 32 | 64 | — | $\sim 1\times 10^{-4}$ | Adam |
| N-BEATS | 512 | 32 | 512 | — | $\sim 1\times 10^{-3}$ | Adam |
| Chronos-T5 | 512 | 32 | 512 | 4096 bins | $\sim 1\times 10^{-3}$ | Adam |
| Lag-Llama | 512 | 32 | 144 | — | $\sim 1\times 10^{-4}$ | Adam |
| NetBurst | 512 | 32 | 512 | 4096 bins | $\sim 1\times 10^{-4}$ | Adam |

## A.3 HYPERPARAMETER DETAILS

Table 5 summarizes all hyperparameters. We use early stopping with a patience of 10 validation evaluations (monitoring validation loss) and select the best checkpoint. For a fair comparison, we fix the context length and batch size across all models; other settings follow each baseline's recommended configuration unless noted.

