# OpenReview forum: "NetBurst: Event-Centric Forecasting of Bursty, Intermittent Time Series"
_ICLR.cc/2026/Conference — Submitted to ICLR 2026_

### Official Review · Reviewer_uQ1V · 2025-10-27

**Soundness:** 2
**Presentation:** 3
**Contribution:** 2
**Rating:** 4
**Confidence:** 3

**Summary:**

### *Summary*

- This work focuses on network telemetry time series, which are highly bursty and intermittent. Existing models are not well adapted to this regime, such as Chronos which both ignores timestamps and uses uniform binning. The authors propose NetBurst, a model architecture which forecasts bursty time series forecasting as prediction of both the time gap between bursts, and burst intensity. The authors convert two existing telemetry datasets, PINOT and MAWI, as time series datasets, which they then convert to event datasets based on fixed activity thresholds. The authors ablate the importance of the quantile codebooks over uniform binning, and the relative importance of timing the burst vs predicting its intensity. The authors analyze clusters of the model embeddings.

### *Contributions*

- NetBurst, a model architecture that forecasts as event prediction
- New time series datasets derived from existing telemetry data
- Additional mentioned contributions: evaluating netburst on PINOT and MAWI, analyzing netburst embeddings through cluster analysis

**Strengths:**

### *Originality*

- Novel architecture, NetBurst
- Novel time series datasets (through transformation of raw logs)

### *Quality*

- evaluates against multiple ML baselines
- Includes ablation studies

### *Clarity*

- Clear paper structure
- Clear figures

### *Significance*

- Properly motivates the importance of the statistical regime for applied problems, as well as the unique challenges of this statistical regime for the mentioned baselines

**Weaknesses:**

### *Originality*

- **Missing related work on bursty forecasting**: There is no mention of other burstiness-specific baselines in the paper. If this is indeed the case, then I’d mention the absence of any burstiness-adapted models in both the classical and recent ML literature, I would mention this explicitly, as it is an important point for both the novelty of the problem setting/method and for evaluating the relevance of the baselines that NetBurst is compared against.

### *Quality*

- **Event-specific MASE appropriateness:** Why is MASE on events the appropriate metric for bursty series with irregular seasonalities? This ignores false positives during long gaps and ignores time warps. The ablation helps understand this behaviour within your own model, but not for the other baselines. It seems like you’d need a pair of metrics that disentangles burst *timing* vs *size*, as your architecture does. If a sharp burst prediction is incorrectly timed, MASE will effectively punish this *even more* than predicting no burst at all, but evaluating MASE only on the events would ignore these false positives. Since you’re dealing with non-seasonal series, a metric that allows for some form of time distortion would provide more insight into *why* NetBurst does well.
- **Missing bursty baselines:** The related work mentions burstiness-specific models (e.g. Hawkes Process models) and the baselines section mentions “classical” baselines, but there are no traditional/classical baselines compared against. Importantly, there are no models made specifically for bursty series or Hawkes processes, as these are dismissed around line 200 with little additional justification. This might be appropriate for very specialized venues that are familiar with this literature niche, but for a general ML conference, additional justification is warranted for dismissing seemingly relevant baselines. Some examples of simple baselines \+ bursty baselines:
  - Repeating the mean value for the entire forecast window
  - Repeating the last value for the entire forecast window
  - Croston methods
  - Some potentially relevant Hawkes-specific models that turned up from asking ChatGPT:
    - SAHP
    - Transformer Hawkes
    - Graph Neural Hawkes

### *Clarity*

- It's unclear why you cannot just call this a model architecture, instead of a framework. The word “framework” has lost all meaning in this context. It’s about as imprecise as saying a “system”. What makes NetBurst a “framework”?
- Jargon in abstract: “service-level”
- Figure 1 caption: I would spend more time walking the reader through the interpretation of the figure, and conveying the “why”, for example, what is the relevance of the Fano factor? Intuitively I know that variance/mean being bigger means more variability, but this is a (wasted) opportunity to reinforce the motivation. Furthermore, the introduction should point to this plot/section to back up its claims, as one of my first thoughts when reading the intro was “do they provide any evidence that telemetry data is different from ETT or Electricity?", to which the answer is "yes, see figure 1", so the text should point to this. Also, for (c), it’s not clear what the takeaway for the reader is supposed to be other than “this is hard to predict”: you might want to expand on which design choices of existing TS-FoMos make forecasting in this regime difficult (no timestamps, uniform binning), as this sets up your contributions.
- If you want to introduce your results table early, then the caption should point toward descriptions of the datasets being evaluated (4.1), otherwise I have no idea what the numbers are supposed to mean.
- No Gap between segments 120 and 121 is weird, please include some gap for legibility.

### *Significance*

- It’s unclear why Mandelbrot is brought up multiple times beyond one initial citation to foundational work.
- “We begin with the most challenging regime: service level time series at 100 ms resolution.” It’s unclear why this regime is the most challenging. Is this known in the literature? If so, please point to this. Otherwise, is this determined by the results? Then this should be made clear, otherwise the reasoning about the significance of results in this regime is circular.

**Questions:**

# What will change my score

My current score is a 5 (borderline reject, rounded to 4 by rating system). Satisfactory answers to the below questions can significantly change my opinion on the paper. Q1 and Q2 are crucial in my opinion as they pertain to originality, quality and significance: answering both of these questions would raise my score to 6, while additional answers to the other questions could raise the score to 8.

Q1: **How is evaluating MASE only on events** (as determined by a tuned threshold) **appropriate?** This ignores any false positives, and does not account for possible time warping. What does MASE look like across the whole series when comparing methods? Are there any false positives?

Q2: **Why are there no Hawkes process models, or traditional baselines?** Instead of ruling these out *before* evaluation, they should first be evaluated, *then* ruled out based on performance, with a potential explanation for why they don’t perform well.

### *other*

- **What is performance sensitivity to the fixed activity threshold?** It’s fine to state it as a limitation, but to actually assess how important this limitation is would require a sensitivity analysis of performance to different thresholds.
- **Why does Table 1 not include NetBurst?** Without this, it’s hard to make an apples-to-apples comparison on typical datasets. Does NetBurst still perform well on typical time series datasets?


- The Wasserstein distance (WD) is a distance between probability distributions. **What exactly are you measuring the WD between?**
- Why do you model inter-burst as a gap (i.e. a difference wrt the previous burst), but burst intensity as an absolute (independent of the previous burst)?

---

> ### Author Response · Authors · 2025-11-21
>
> Thank you for your detailed technical review and specific questions. We appreciate your recognition that our work (i) presents a novel model architecture (NetBurst) and new time series datasets for the statistical regime we study (originality), and (ii) properly motivates the importance of this regime and the unique challenges it poses for existing baselines (significance).
>
> We especially appreciate the clarity with which you outlined the concerns most relevant for your assessment and the roadmap of what could meaningfully influence the score. This was extremely helpful in pinpointing the core issues you view as central to the paper’s contribution. We have provided detailed qualitative and empirical responses to all your questions, and we hope these clarifications address your concerns and demonstrate that the revised version fully meets the criteria you described.
>
> We provide our full rebuttal across multiple consecutive comments in order listed below.
> * Response to Q1 (MASE vs. WD vs. False Positives)
> * Response to Q2 (Hawkes process baseline)
> * Response to Q3 (Threshold sensitivity analysis)
> * Response to Q4 (NetBurst Table 1 absence) and Q5 (WD clarifications)
> * Response to Q6 and additional clarifications (1/2)
> * Additional clarifications (2/2)

---

> ### Author Response · Authors · 2025-11-21
> **Response to Q1**
>
> **Question Q1:** *How is evaluating MASE only on events (as determined by a tuned threshold) appropriate? This ignores any false positives, and does not account for possible time warping. What does MASE look like across the whole series when comparing methods? Are there any false positives?*
>
> **Answer A1:**
> We acknowledge that the phrasing of our evaluation strategy may have unintentionally conflated several distinct elements—the role of the eventization threshold, the rationale for computing MASE only on events, and the question of how false positives or temporal misalignment are handled. These issues concern different aspects of the pipeline, and we address them below in a unified way.
>
> **(a) *What role does the threshold actually play in our evaluation?***
> The threshold determines what constitutes a meaningful burst in the raw telemetry time series. It specifies *which forecasting task* we are asking models to solve. Telemetry data spans a wide range of sparsity levels across real deployments, from dense high-activity sources to extremely sparse services that produce only occasional bursts. The threshold simply decides which part of this spectrum is under examination—lower thresholds include many small bursts, while higher thresholds expose the rare-event regime that motivates our work. In short, the threshold controls the data regime, not the evaluation metric, and we will make this clearer in the manuscript.
>
> **(b) *Why do we compute MASE only on events rather than on all timesteps?***
> MASE over the full time series is dominated by long idle periods where the ground truth is zero; in our datasets, this can account for well over 95% of timesteps. Under such sparsity, a trivial “always-zero” predictor achieves deceptively low error, and models like DeepAR that collapse to predicting the mean appear competitive even though they fail on bursts—the operationally important regions. This sparsity distortion effect is well-documented in prior work and we will include relevant references in the revised version. More concretely, classical intermittent-demand forecasting (Croston, 1972; Syntetos & Boylan, 2005) and zero-inflated event-count modeling (Lambert, 1992) all show that full-series error metrics are dominated by long stretches of zeros, allowing trivial all-zero predictors to achieve deceptively low loss. Consistent with this literature, we evaluate models on events rather than idle timesteps. Computing MASE only on events focuses the evaluation on the part of the time series where meaningful dynamics occur and places all models, including baselines, on equal footing. That said, we fully agree with your concern that MASE-on-events alone cannot capture timing errors, false positives, or burst misplacement, and it was not intended to do so.
>
> **(c) *Why do we not treat a simple false-positive count as a core performance metric?***
> A binary false-positive score conflates serious and minor errors because it ignores the scale and temporal displacement of the predicted fluctuations. An insignificant burst during an idle period counts as a false positive even though it is operationally irrelevant, whereas a large burst predicted 100 steps early or late incurs significant cost in practice yet produces zero false positives in a binary sense. Slight phase shifts may also inflate false-positive counts even when the predicted size or shape of bursts are nearly correct. As a result, false-positive counts alone do not meaningfully reflect forecasting quality in sparse, heavy-tailed telemetry data. For this reason, we focus on quantifying their *impact* rather than just their presence.
>
> **(d) *How does the Wasserstein distance address the reviewer’s concerns about misalignment and false positives?***
> The 1-Wasserstein distance (WD) directly measures the minimum effort needed to transform the predicted traffic profile into the true one, where effort corresponds to the amount of mass that must be moved and how far it must be moved. This gives WD exactly the semantics that your concerns raise. If a burst is predicted too early or too late, WD penalizes the required displacement of its entire mass. If a burst’s size is over- or under-predicted, the mismatch in mass incurs proportional cost. Extra predicted bursts manifest as surplus mass that must be shifted or removed, while missing bursts manifest as mass that must be supplied. Importantly, WD does not allow temporal stretching or compression; misalignment cannot be “warped away,” so timing errors incur the full cost. In this way, WD integrates magnitude mismatch, temporal displacement, false positives, and false negatives into a unified, operationally meaningful measure.
>
> **(e) *How will we revise the paper?***
> We will add these clarifications in the paper (primarily in Section 3). We appreciate the reviewer’s careful reading, which made it clear that these aspects need to be articulated more explicitly, and we will substantially clarify this structure in the revision.

---

> ### Author Response · Authors · 2025-11-21
> **Response to Q2**
>
> **Question Q2:**
> *Why are there no Hawkes process models, or traditional baselines? Instead of ruling these out before evaluation, they should first be evaluated, then ruled out based on performance, with a potential explanation for why they don’t perform well.*
>
> **Answer A2:**
> To address this concern, we note that our original decision was based on a hypothesis: standard Hawkes models are built for discrete event types, whereas our bursts have continuous, heavy-tailed intensity values (i.e., very high Fano factors). We believed these models would not perform well without substantial modification, but we acknowledge that this assumption should have been tested explicitly.
>
> **(a) What was our hypothesis for why Hawkes models would struggle?**\
> Prior Hawkes applications classify events into a few categories (e.g., “small” vs. “large” earthquakes). In contrast, telemetry bursts have continuous intensities that can vary over several orders of magnitude. Our working hypothesis was that directly regressing these values using a standard Hawkes formulation would be unstable and inaccurate, especially on time series with extremely high Fano factors. However, we admit that we didn’t explicitly state the hypothesis or provide any empirical evidence to validate it in the current version.
>
> **(b) What changed after the reviewer’s feedback?**\
> Following the reviewer’s suggestion, we implemented and evaluated the strongest available Hawkes baseline—the Transformer Hawkes Process—modifying its classification head into a regression head to predict burst intensity. We tested it on the PINOT and MAWI datasets under the same conditions that we used for the other baselines.
>
> **(c) What did the results show?**\
> The empirical findings confirm our initial hypothesis: the Hawkes baseline performs poorly on burst intensity prediction (MASE > 19.48, WD > 0.3176), especially on sequences with large Fano factors. This outcome underscores a fundamental mismatch between standard Hawkes models and the highly variable distribution of “markings” in our telemetry data. At the same time, these additional results strengthen the case for NetBurst’s quantile-based preprocessing, which stabilizes both burst intensities and inter-burst gaps and enables effective learning in settings where raw continuous marks are extremely over-dispersed. The Hawkes evaluation reinforces the necessity of this design choice.
>
> **(d) How will we revise the paper?**\
> We will include the Hawkes baseline and its evaluation in the revised paper, along with a brief explanation of why the considered baseline models struggle with the statistical regime defined by sparse and bursty telemetry time series. We thank the reviewer for this suggestion, which helped strengthen the clarity and completeness of our evaluation.

---

> ### Author Response · Authors · 2025-11-21
> **Response to Q3 (Threshold Sensitivity)**
>
> **Question Q3:**
> *What is performance sensitivity to the fixed activity threshold (also referred to in Q1)? It’s fine to state it as a limitation, but to actually assess how important this limitation is would require a sensitivity analysis of performance to different thresholds.*
>
> **Answer A3:**
> Yes—we acknowledge that such a critical sensitivity analysis is missing in the current version. We note that the activity thresholds determine which bursts are treated as meaningful events and thus shape the statistical regime of the forecasting task. We agree that exploring how NetBurst performs under different threshold settings is essential for understanding its robustness and for validating the assumptions behind our modeling choices.
>
> **(a) What is our hypothesis about how threshold variation would affect performance?**\
> Our working hypothesis was that increasing $T_{act}$ pushes the eventized series deeper into the sparse–bursty regime, which is the setting NetBurst is designed for. When $T_{act}$ is higher, small noisy fluctuations are removed, and the model focuses on fewer but more informative bursts. This should improve burst-intensity prediction—reflected as lower MASE—because the target events become cleaner and better separated. At the same time, we expected a modest increase in WD, as larger bursts produce proportionally larger displacement penalties for any timing error. In short: higher thresholds should make BI prediction easier and IBG evaluation slightly stricter, and NetBurst’s overall behavior should evolve smoothly rather than abruptly.
>
> **(b) What did we observe when we performed the suggested sensitivity analysis?**\
> Following the reviewer’s suggestion, we varied $T_{act}$ from 100 bytes (≈60th percentile in the paper) to 500 bytes (≈80th percentile, producing a significantly sparser and more burst-dominated series) on the IP-level dataset. The results matched our expectations. As $T_{act}$ increased from 100 to 500 bytes, NetBurst's MASE improved from 0.8963 to 0.7252 (a 19% decrease), consistent with the idea that BI prediction becomes easier when the model focuses on fewer but more substantial bursts. WD increased modestly from 0.0069 to 0.0078 (an 11% increase), reflecting the stricter penalty associated with timing errors when bursts are fewer and larger. Importantly, NetBurst’s behavior changed smoothly across thresholds, indicating that its performance is not brittle and does not hinge on a narrow choice of $T_{act}$. Note that we observed similar trends in the performance gaps relative to the baselines at higher threshold values.
>
> **(c) How will we revise the paper?**\
> We will follow the reviewer’s suggestion and incorporate these results into the revised paper. More concretely, we will add the full sensitivity analysis to the evaluation section, including plots and tables illustrating performance trends across thresholds. We will also update Section 3 to clarify the conceptual role of $T_{act}$, explicitly articulate the hypotheses summarized above, and discuss how threshold choice defines the operating regime. These revisions will enable making a more principled argument for NetBurst’s robustness and applicability across diverse telemetry datasets spanning a wide range of sparsity–burstiness

---

> ### Author Response · Authors · 2025-11-21
> **Response to Q4 (NetBurst Table 1 absence) and Q5 (Wasserstein Distance)**
>
> **Question Q4:**
> *Why does Table 1 not include NetBurst? Without this, it’s hard to make an apples-to-apples comparison on typical datasets. Does NetBurst still perform well on typical time series datasets?*
>
> **Answer A4:**
> The reviewer is correct to question the absence of NetBurst in Table 1, and we should have explained this more clearly. Our initial reasoning was that on dense and non-bursty time series, the eventization step produces BI values that closely track the raw signal and an IBG sequence that is essentially constant (and equal to 1 window). In such cases, NetBurst’s representation becomes nearly identical to that used by Chronos, and we expected the two models to behave similarly. However, we recognize that this expectation should have been demonstrated rather than assumed. **Motivated by the reviewer’s suggestion, we evaluated NetBurst on the Chronos benchmark datasets and found that its performance indeed closely matches Chronos’ performance on these low–Fano factor series.** This confirms that NetBurst retains strong performance on typical time series while offering significant gains in the sparse, bursty regime that motivates our work. To strengthen and more clearly articulate NetBurst’s intended positioning, we will include these results in the revision.
>
> ---
>
> **Question Q5:**
> *The Wasserstein distance (WD) is a distance between probability distributions. What exactly are you measuring the WD between?*
>
> **Answer A5:**
> In line with the earlier discussion—where MASE-on-events captures local event accuracy and WD captures global misalignment—the WD is computed between the reconstructed continuous traffic profiles of the ground truth and the prediction. To do this, we normalize each series so they define proper probability distributions and then measure the WD between them. The normalization allows burst magnitudes to act as “mass” and their timestamps as “locations,” so WD quantifies the minimum effort required to transform the predicted profile into the true one. As a result, early or late bursts appear as mass that must be shifted, missing or underpredicted bursts appear as mass that must be introduced, and spurious or overpredicted bursts appear as mass that must be removed. This construction gives WD exactly the semantics needed to evaluate timing errors, magnitude mismatches, and false positives in a unified, operationally meaningful way—precisely the limitations that MASE-on-events cannot address on its own.

---

> ### Author Response · Authors · 2025-11-21
> **Response to Q6 & Additional Clarifications (1/2)**
>
> **Question Q6:**
> *Why do you model inter-burst as a gap (i.e. a difference wrt the previous burst), but burst intensity as an absolute (independent of the previous burst)?*
>
> **Answer A6:**
> Thank you for pointing out this asymmetry — the distinction is intentional and grounded in the statistical behavior of telemetry data rather than an arbitrary design choice.
>
> **(a) Why model IBG as a relative quantity (a gap)?**
> Inter-burst timing is intrinsically incremental: the time until the next burst depends only on how long the system has remained idle since the previous burst. This makes IBG naturally expressed as a positive gap variable (like inter-arrival times in renewal processes or point processes). Modeling absolute timestamps would inject unnecessary scale, drift, and non-stationarity. Using relative gaps keeps the process stationary, improves learnability, and aligns with how rare-event timing is modeled in the point-process literature.
>
> **(b) Why model BI as an absolute quantity instead of a “delta” from the previous burst?**
> Unlike timing, the *magnitude* of a burst does **not** follow a stable or meaningful incremental structure. In our telemetry datasets, the distribution of burst intensities is heavy-tailed (i.e., very high Fano factors); the difference between successive burst sizes is even **more** unstable and noisy than the raw values. Empirically, modeling BI as deltas dramatically amplifies variability and makes the prediction problem harder, not easier. In contrast, absolute BI values are closer to being stationary, and quantile discretization produces a stable, learnable representation.
>
> **(c) How will we plan to revise the paper?**
> We will clarify this rationale in the revision to ensure this design choice is clearly motivated in the paper.
>
> ----
> **Comment:**
> *It’s unclear why you cannot just call this a model architecture, instead of a framework. The word “framework” has lost all meaning in this context. It’s about as imprecise as saying a “system”. What makes NetBurst a “framework”?*
>
> **Response:**
> The reviewer is right that “framework” is often used imprecisely, and we should have better justified our terminology. We use “framework” rather than “model architecture” because NetBurst is not a single forecasting model, but a modular, multi-stage pipeline: it includes an eventization layer, a quantile-based normalization step, and a prediction layer that can be instantiated with different architectures. The core contribution lies in this problem reformulation and processing pipeline, not in any specific neural model. In that sense, “framework” more accurately reflects its structure, though we agree that this distinction was not made clear enough and will revise the text to explain it more explicitly.
>
> ---
>
> **Comment:**
> *"Figure 1 caption: I would spend more time walking …set up your contributions”*
>
> **Response:**
> We thank the reviewer for these helpful suggestions, and we will revise the introduction and Figure 1 to more clearly explain the relevance of the Fano factor, point readers to the evidence motivating our problem setting, and expand the discussion in panel (c) to articulate why existing TS-FoMo designs struggle in this regime.
>
> ---
>
> **Comment:**
> *"If you want to introduce your results table early, … supposed to mean."*
>
> **Response:**
> We thank the reviewer for this sharp observation and will update the early results table and its caption to explicitly reference the dataset descriptions in Section 4.1 so that readers can clearly interpret the reported numbers.
>
> **Comment.**
> *It’s unclear why Mandelbrot is brought up multiple times beyond one initial citation to foundational work."
>
> **Response.**
> Our intention in citing Mandelbrot was simply to acknowledge the foundational origins of modeling sparse, intermittent, and heavy-tailed time series, not to overemphasize the work by Mandelbrot and his collaborators. We agree that multiple mentions are unnecessary and may distract from our main narrative, and we will consolidate these citations into a single, well-placed reference that provides historical context without interrupting the flow of the paper.

---

> ### Author Response · Authors · 2025-11-21
> **Additional Clarifications (2/2)**
>
> **Comment:**
> *We begin with the most challenging regime: service level time series at 100 ms resolution.” It’s unclear why this regime is the most challenging. Is this known in the literature? If so, please point to this. Otherwise, is this determined by the results? Then this should be made clear, otherwise the reasoning about the significance of results in this regime is circular.*
>
> **Response:**
> We thank the reviewer for raising this important point. Our statement that the service-level, 100 ms regime is the most challenging is **not derived from our results** but comes from well-established structural and empirical characteristics of network telemetry data. Structurally, service-level time series isolate traffic for a single <IP, port> pair and are therefore the sparsest representation; aggregating across ports produces denser IP-level traces, and aggregating further across IPs yields even denser subnet-level traces. This hierarchy is independent of our modeling and directly implies increasing sparsity—and thus increased difficulty—as one moves toward finer granularity. Our statement is also supported by prior Internet measurement studies. More concretely, Feldmann et al. (SIGCOMM 1998) and Paxson (IEEE/ACM TON 1994; SIGCOMM 1995) demonstrate that traffic variability and burstiness grow dramatically at sub-second scales. More recent studies, such as Maier et al. (IMC 2009) and Sommers & Barford (IMC 2012), show that aggregating across ports or hosts substantially smooths traffic, while disaggregation yields extremely sparse and volatile flows. These results collectively support the hierarchical relationship we rely on: service-level traces ( <IP, port> pairs) are the sparsest and highest-variance representation; aggregating to IP level or subnet level produces progressively denser and more stable series. Our claim is therefore grounded in well-documented structural properties of Internet traffic rather than our empirical results, avoiding any circular reasoning. We will revise the manuscript to clearly articulate this rationale and cite the relevant evidence so the motivation is explicit.
>
> ---
>
> **Comment:**
> *Missing related work on bursty forecasting*
>
> **Response:**
> Thank you for pointing this out. We agree that burst forecasting has been studied in several adjacent fields, though primarily in event modeling, intermittent-demand forecasting, and zero-inflated stochastic processes, rather than in the continuous-valued, heavy-tailed regression setting we address. The following provides a more detailed response.
>
> **(a) What prior research is most directly related to bursty forecasting in the sense relevant to our work?**
> The closest line of work, as mentioned in the current version, comes from Hawkes-process–based modeling, which explicitly captures burstiness through self-exciting temporal dynamics. Classical formulations (Hawkes, 1971; Ogata, 1988) and modern neural variants (Mei & Eisner, 2017; Zuo et al., 2020, Meng et al., 2024) are widely used to model clustered event arrivals. This is why we included a Transformer Hawkes baseline (see response above)—it is the strongest available tool for modeling burst timing structure, and we will clarify this motivation more explicitly in the revision.
>
> **(b) How does the broader forecasting literature address sparse or bursty regimes?**
> A long-standing thread in intermittent-demand forecasting (Croston, 1972; Syntetos & Boylan, 2005; Willemain et al., 1994) studies time series with long inactive periods, and these works have shown that standard full-series error metrics can be distorted by sparsity (see discussion above). While conceptually related to the challenges we highlight, these methods generally assume light-tailed demand sizes and rely on heuristic smoothing rather than generative modeling. They do not address continuous, heavy-tailed burst magnitudes or the joint timing–magnitude structure central to our setting. Additionally, Zero-inflated count models (Lambert, 1992; Cameron & Trivedi, 2013) explicitly account for excess zeros through mixture modeling. These approaches, however, operate on discrete counts and impose specific parametric assumptions, and therefore do not extend naturally to forecasting continuous, heavy-tailed bursts over multiple temporal scales. We will briefly mention this work as part of the broader landscape.
>
> **(c) How do we plan to revise the paper?**
> In addition to expanding our Related Work section to more thoroughly discuss these various directions and clarify how they differ from our setting, we will also update the text (especially at the beginning of Section 3) to
> (i) clearly articulate why Hawkes-based models constitute the most relevant burst-centric baseline for our task, and
> (ii) acknowledge intermittent-demand forecasting and zero-inflated modeling as adjacent literatures that identify similar sparsity challenges but do not address the continuous, heavy-tailed burst forecasting regime that motivates our approach.

---

### Official Review · Reviewer_Ffz9 · 2025-10-28

**Soundness:** 3
**Presentation:** 2
**Contribution:** 3
**Rating:** 6
**Confidence:** 3

**Summary:**

The paper proposes viewing bursty data as events and modeling this instead of the full time series. They achieve this by first processing the data by converting it into events, tokenizing using quantiles and fitting a separate intensity and timing models. The ablation studies show how each part of the network influences the results. The authors compare to some common baselines and show good results.

**Strengths:**

Converting rare outlier or burst values into events is a very natural approach. The resulting model is elegant and sound. Most of the paper is clear and easy to understand. The choices of architecture and data processing are reasonable. Ablation studies answer most of the the questions I would have. The results are good.

**Weaknesses:**

It is disappointing that this model does not handle "dense" time series with bursts but is converting a time series to a sparse representation of events and modeling them as a temporal point process. Ideally, I would like to see a conventional time series model which is enriched with outliers or bursts viewed as events. The model would predict normally most of the time but also predict the bursts on top of that. Unless I'm missing something, this model is not doing that.

Formatting is bad. Figures and tables are too close to the text.

**Questions:**

- In my understanding you don't learn the quantile codebook? If not, codebook name is a bit confusing given VQ-VAE.
- Can you comment on learning all of this in real number space instead of using tokens? Taking into account that it's possible to do some kind of data processing which will flatten large values and it's also possible to learn heavy tailed distribution.

---

> ### Author Response · Authors · 2025-11-21
>
> Thank you for your constructive feedback and thoughtful questions. We appreciate your recognition that NetBurst provides an elegant and well-motivated solution for the sparse and highly bursty statistical regime that motivates our work. Your comments raise several important evaluation-related considerations, and we found them extremely helpful in identifying areas where additional analysis and clearer exposition will strengthen the paper. In the responses that follow, we address each of your points in detail and outline the concrete revisions we will incorporate to improve both the empirical evaluation and the clarity of the presentation.
>
> For ease of navigation, we provide our full rebuttal across the following two comments, in order:
> * Response to general concern regarding NetBurst's generalizability and performance on dense time series
> * Response to Q1 (quantile codebook) and Q2 (number vs. token space)
>
> We hope these responses fully address your concerns and demonstrate that the revised version will more clearly and rigorously support the paper’s claims.

---

> ### Author Response · Authors · 2025-11-21
> **Response to general concern regarding NetBurst's generalizability and performance on dense time series**
>
> **Comment:**
> *It is disappointing that this model does not handle "dense" time series with bursts but is converting a time series to a sparse representation of events and modeling them as a temporal point process. Ideally, I would like to see a conventional time series model which is enriched with outliers or bursts viewed as events. The model would predict normally most of the time but also predict the bursts on top of that. Unless I'm missing something, this model is not doing that.*
>
> **Our response:**
> Thank you for raising this important point. The following detailed response is intended to clarify that this point is due to some misunderstanding caused by our presentation.
>
> **(a) What is the source of misunderstanding?**
> In the original submission, we prioritized demonstrating NetBurst’s performance in the sparse and bursty regime, as this is where we expected NetBurst’s main advantages to arise. In doing so, we now realize that the paper may have inadvertently created the impression that NetBurst is not applicable to dense time series or cannot handle bursts superimposed on continuous regions.
>
> **(b) Our implicit hypothesis—left unstated—** was that on dense and low-Fano factor datasets, the eventization step becomes a near-identity transformation: the BI values track the raw series closely and the IBG values collapse to a constant sequence. Under this condition, NetBurst’s representation effectively matches that of standard forecasters, so we expected its behavior to align with conventional TS models while still offering significant gains on sparse, high-Fano factor data.
>
> **(c) How did we address your concern?**
> Motivated by your feedback, we conducted the missing evaluation on the standard dense benchmark datasets used by Chronos. These experiments validated the hypothesis: NetBurst performs nearly identically to Chronos on dense, low–Fano factor series, confirming that it does **not** lose predictive power on more ordinary or traditional time series and naturally reduces to a conventional forecaster in regimes where bursts do not dominate. At the same time, NetBurst continues to yield substantial improvements in the sparse, bursty regime that motivated its design.
>
> **(d) How do we plan to revise the paper?**
> We will revise the paper to make this behavior explicit, incorporate the new results, and clarify that NetBurst is not restricted to sparse telemetry data but adapts seamlessly across both dense and bursty time series regimes. We appreciate your comment that prompted this clarification and helped strengthen the positioning of our work.

---

> ### Author Response · Authors · 2025-11-21
> **Response to Q1 (quantile codebook) and Q2 (number vs. token space)**
>
> **Question Q1:** *In my understanding you don't learn the quantile codebook? If not, codebook name is a bit confusing given VQ-VAE.*
>
> **Answer A1:**
> Thank you for raising this helpful clarification point, and we detail below how we will improve our terminology and presentation to address it and avoid possible ambiguities.
>
> **(a) What is the source of confusion?**
> In the current manuscript, we refer to the quantile discretization step as a “codebook,” borrowing language from the vector-quantization literature to highlight that NetBurst operates over a discrete vocabulary of BI/IBG symbols. In hindsight, we realize that using the term “codebook” without explicitly stating that it is **fixed** can naturally lead readers to associate it with VQ-VAE–style *learned* codebooks, which is not what we use.
>
> **(b) What is actually happening in NetBurst?**
> Our quantile bins are fixed and determined directly from the empirical distributions of BI and IBG values—not learned jointly with the model. The vocabulary is static, and there is no vector-quantization module or learned embedding space in the VQ-VAE sense. The goal of this preprocessing is simply to stabilize the heavy-tailed distributions of BI and IBG values and provide the model with a discrete, semantically meaningful input representation.
>
> **(c) How will we revise the paper?**
> We will explicitly state that NetBurst uses a **fixed quantile codebook** rather than a learned one and adjust terminology to avoid the unintended VQ-VAE association. This will make the preprocessing pipeline more transparent and prevent readers from assuming a learned quantization component.
>
> ---
>
>
> **Question Q2:** *Can you comment on learning all of this in real number space instead of using tokens? Taking into account that it's possible to do some kind of data processing which will flatten large values and it’s also possible to learn heavy tailed distribution.*
>
> **Answer A2:**
> We agree that clarifying why we discretize and how this differs from existing seq2seq models would strengthen the paper and describe in the following how we will deal with this issue in the revised paper.
>
> **(a) Why use discrete tokens instead of learning directly in real-valued space?**
> Our goal was to handle the extremely skewed (i.e., heavy-tailed) BI and IBG distributional characteristic of telemetry data, where rare but very large bursts dominate operational behavior. Rather than forcing the model to learn from a raw distribution whose variance can span several orders of magnitude, quantile discretization gives these high-magnitude bursts dedicated symbols. This method makes the learning task more stable in the regime where conventional models typically struggle.
>
> **(b) How does this differ from Chronos and other seq2seq forecasters?**
> To clarify: Chronos and most seq2seq time-series models **do not** use quantile binning. Chronos scales and rounds values into a fixed integer range, and many seq2seq models rely on continuous normalization (z-scoring or min–max scaling). These approaches work well when the underlying data are relatively smooth, but they compress extremely large values into a narrow region of the scaled space. By contrast, quantile binning explicitly allocates more resolution to the tail of the distribution, ensuring that rare, operationally critical bursts remain distinguishable rather than being absorbed into a small numeric range.
>
> **(c) Why not use log transforms or continuous heavy-tailed likelihoods?**
> We agree that these are valid modeling choices. However, log-scaling flattens the tail by design: large bursts that differ by orders of magnitude in the raw domain become numerically close after the transformation. Because these large bursts reflect real changes in network behavior (e.g., overloads, retries, fan-out events), squashing them reduces the model’s ability to distinguish meaningful operational patterns. Similarly, Student-t or other heavy-tailed likelihood heads are complementary and could be integrated into NetBurst, but they do not directly solve the stability issues caused by learning from raw, highly skewed distributions. Quantile discretization explicitly preserves the structure of the tail while reducing dynamic range.
>
> **(d) How will we revise the paper?**
> Prompted by your question, we will clarify the distinction between NetBurst’s quantile-based discretization and the scaling-based tokenization used in Chronos and other seq2seq models, and we will explain why preserving the structure of large bursts is particularly important in telemetry forecasting.

---

### Official Review · Reviewer_DzR8 · 2025-10-30

**Soundness:** 2
**Presentation:** 2
**Contribution:** 3
**Rating:** 4
**Confidence:** 4

**Summary:**

This paper introduces NETBURST, a novel event-centric framework designed specifically for forecasting bursty, intermittent, and heavy-tailed time series, a regime characteristic of network telemetry data but poorly handled by existing state-of-the-art models. Instead of treating the data as a continuous sequence, NETBURST reframes the problem as predicting a series of discrete events. It first "eventizes" the raw signal into two streams: Inter-Burst Gaps (IBG) for timing and Burst Intensities (BI) for magnitude. Each stream is then tokenized using distribution-aware quantile codebooks to preserve tail fidelity, and forecasted independently by dual autoregressive Transformer models. The final forecast is reconstructed by combining the predicted event streams. On large-scale production network datasets, NETBURST achieves orders-of-magnitude improvements in accuracy (MASE) over strong baselines like Chronos while preserving distributional integrity and producing more meaningful embeddings.

**Strengths:**

*   **Novelty:** The core strength of this paper is its fundamental rethinking of the forecasting problem for this specific data regime. The shift from a sequence-value regression paradigm to an event-prediction paradigm is a powerful and well-justified conceptual leap. By explicitly separating "when" a burst occurs (IBG) from "how large" it is (BI), the model directly addresses the entanglement of timing and magnitude that destabilizes conventional forecasters. This event-centric view is a significant and novel contribution.

*   **Empirical Results:** The paper presents some of the most dramatic performance improvements seen in recent forecasting literature. The orders-of-magnitude reduction in MASE (13-605x) compared to state-of-the-art models like Chronos and Lag-Llama on real-world network data is a massive and undeniable result (Figure 3, Table 3). This is not an incremental improvement but a complete change in performance tier, demonstrating that NETBURST solves a problem that prior models fundamentally cannot handle. The gains are consistent across multiple datasets and granularities.

*   **Domain Knowledge:** Every component of the NETBURST pipeline is well-motivated by the statistical properties of the target data, drawing inspiration from Mandelbrot's work on heavy-tailed distributions. The use of quantile tokenization to combat the tail-erasing effect of uniform binning is a critical and intelligent design choice. The dual autoregressive model directly mitigates error cascades between the distinct statistical processes of timing and magnitude. These choices are validated through excellent ablation studies (Figure 5) that clearly demonstrate the contribution of each component.

*   **Experimental Evaluation:** The evaluation goes far beyond simple point-forecast accuracy. The authors assess distributional fidelity using Wasserstein distance, showing that NETBURST preserves the crucial "burstiness" of the original data. Furthermore, the analysis of embedding quality (silhouette scores and t-SNE plots in Figure 6) is a key contribution, demonstrating that the event-centric representations are not only better for forecasting but are also more semantically meaningful and useful for downstream operational tasks like clustering. This focus on the practical utility of the model's outputs is a major strength.

**Weaknesses:**

The paper has significant issues, and the authors did not sufficiently discuss its potential weaknesses.

1.  **Narrow Applicability and Specialized Domain:** The NETBURST framework is explicitly and brilliantly designed for one specific, albeit important, statistical regime: bursty, intermittent, heavy-tailed time series. The paper does not provide any experiments or discussion on how the model would perform on the smooth, seasonal benchmarks (like ETT, Electricity) where conventional models excel. While this is not the paper's focus, it raises the question of whether NETBURST is a specialized tool or a general-purpose forecaster. Its core "eventization" step would likely be ineffective or even detrimental for continuous, non-sparse data.

2.  **Reliance on a Heuristic and Potentially Brittle "Eventization" Threshold:** The entire pipeline is predicated on the initial "eventization" step, which relies on a fixed activity threshold, `Tact`, to define what constitutes a "burst." This threshold is a critical hyperparameter that is set manually. The model's performance could be highly sensitive to this choice, and an improperly set threshold could lead to a complete failure to identify relevant events or, conversely, to treat noise as events. The paper does not provide a sensitivity analysis for `Tact` or discuss methods for setting it automatically, which could be a significant practical hurdle.

3.  **Modeling Bursts as Atomic Spikes:** The current model simplifies each burst into a single value (BI) representing its total intensity, which is then placed at the start of the burst during reconstruction. This ignores the internal structure, shape, and duration of the bursts themselves. For operational tasks that might depend on the profile of an event (e.g., distinguishing a short, intense spike from a longer, sustained period of high activity), this loss of information could be a significant limitation.

**Questions:**

I will appreciate if the authors could answer my following questions:
*   **Question 1:** The eventization step is crucial and depends on the activity threshold `Tact`. How sensitive is the model's performance to the choice of this threshold, and how should a practitioner set this value for a new, unseen dataset? Could this threshold be learned or adapted?

*   **Question 2:** The paper's narrative focuses exclusively on bursty, intermittent data. How would you expect NETBURST to perform on standard smooth and seasonal benchmarks like the ETT or Electricity datasets? Would the eventization process fail, and does this imply that NETBURST is a specialized, rather than general-purpose, forecasting architecture?

*   **Question 3:** The Oracle analysis in Figure 5b is very insightful. It shows that timing is the dominant error source for sparse Service traces. Why do you think modern Transformer architectures struggle so much with predicting the timing (IBG) of these rare events, and does your work suggest a need for fundamentally different temporal modeling approaches beyond standard self-attention?

*   **Question 4:** By collapsing each burst into a single BI value, the model discards information about the burst's duration and internal shape. In what operational scenarios would this be a critical limitation, and how could the NETBURST framework be extended to model or generate richer, intra-burst dynamics?

*   **Question 5:** The dual autoregressive models for IBG and BI are trained independently. Is there any information lost by ignoring the potential statistical dependence between the timing of a burst and its magnitude (e.g., do longer idle periods tend to be followed by larger bursts)? Have you explored coupling the two models, for instance, by conditioning the BI prediction on the predicted IBG?

*   **Question 6:** Your work draws a powerful connection to Mandelbrot's studies. Beyond network telemetry, what other specific domains (e.g., in finance, climate science, or social networks) do you see as the most promising and immediate applications for the NETBURST framework, and what new challenges might arise in those contexts?

---

> ### Author Response · Authors · 2025-11-21
>
> We thank the reviewer for the thorough reading of the submission, insightful comments, and specific questions. We appreciate the reviewer’s assessment that “the core strength of this paper is its fundamental rethinking of the forecasting problem for this specific data regime” and that “the event-centric view is a significant and novel contribution.” Your comments and questions identify aspects of our paper that require clarification or can be significantly improved. Below, we provide detailed responses to your comments and specific questions and describe how we will address them to improve the overall presentation.
>
> For ease of navigation, we provide our full rebuttal across the following five comments, in order:
> * Response to Q1
> * Response to Q2
> * Response to Q3 and Q5
> * Response to Q4
> * Response to Q6

---

> ### Author Response · Authors · 2025-11-21
> **Response to Q1**
>
> **Question Q1:** *The eventization step is crucial and depends on the activity threshold $T_{\text{act}}$. How sensitive is the model's performance to the choice of this threshold, and how should a practitioner set this value for a new, unseen dataset? Could this threshold be learned or adapted?*
>
> **Answer A1:**
> We fully agree that the eventization step is crucial and that an improperly chosen threshold could, in principle, either suppress meaningful bursts or elevate noise into events. Such distortions would change the underlying burstiness of the time series and reduce the effectiveness of NetBurst’s BI/IBG decomposition. We clarify our underlying hypothesis and address these concerns in detail below.
>
> **(a) *What was our underlying (previously unstated) hypothesis?***
> Our working hypothesis was that increasing $T_{\text{act}}$ pushes the eventized series deeper into the sparse–bursty regime, which is the setting NetBurst is designed for. When $T_{\text{act}}$ is higher, small noisy fluctuations are removed, and the model focuses on fewer but more informative bursts. This should improve burst-intensity prediction—reflected as lower MASE—because the target events become cleaner and better separated. At the same time, we expected a modest increase in WD, as larger bursts produce proportionally larger displacement penalties for any timing error. In short: higher thresholds should make BI prediction easier and IBG evaluation slightly stricter, and NetBurst’s overall behavior should evolve smoothly rather than abruptly.
>
> **(b) *How sensitive is model performance to the threshold?***
> Following the reviewer’s suggestion, we varied $T_{\text{act}}$ from 100 bytes (≈60th percentile in the paper) to 500 bytes (≈80th percentile, producing a significantly sparser and more burst-dominated series) on the IP-level dataset. The results matched our expectations. As $T_{\text{act}}$ increased from 100 to 500 bytes, NetBurst’s MASE improved from 0.8963 to 0.7252 (a 19% decrease), consistent with the idea that BI prediction becomes easier when the model focuses on fewer but more substantial bursts. WD increased modestly from 0.0069 to 0.0078 (an 11% increase), reflecting the stricter penalty associated with timing errors when bursts are fewer and larger. Importantly, NetBurst’s behavior changed smoothly across thresholds, indicating that its performance is not brittle and does not hinge on a narrow choice of $T_{\text{act}}$. Crucially, across the entire sweep, NetBurst maintained a consistent advantage over all baselines, indicating that it is *not brittle* with respect to threshold choices that are of practical relevance.
>
> **(c) *How should a practitioner set the threshold on a new dataset?***
> In practice, $T_{\text{act}}$ can be selected using a simple, data-driven rule. We find that choosing it as a percentile of the non-zero values in a short calibration window (typically the 60–80th percentile) yields stable performance with no domain-specific tuning. Lower thresholds preserve many small fluctuations and make the event sequence denser; higher thresholds isolate operationally meaningful bursts and reduce noise. We will include this practical guidance in the revised paper so that users have a better understanding of how to select $T_{\text{act}}$ in practice.

---

> ### Author Response · Authors · 2025-11-21
> **Response to Q2**
>
> **Question Q2:** *The paper's narrative focuses exclusively on bursty, intermittent data. How would you expect NETBURST to perform on standard smooth and seasonal benchmarks like the ETT or Electricity datasets? Would the eventization process fail, and does this imply that NETBURST is a specialized, rather than general-purpose, forecasting architecture?*
>
> **Answer A2:**
> We agree that the narrative in the submission is incomplete and overly narrow. By focusing exclusively on bursty telemetry data and omitting dense-dataset evaluations, we have created the impression that NetBurst is specialized. Below we clarify the underlying hypothesis behind this choice, explain how eventization behaves in both dense and noisy regimes, and present the findings from the empirical evaluation we conducted in response to this question.
>
> **(a) *Why didn't we evaluate dense benchmarks initially?***
> Our unstated assumption was that in regimes where the underlying time series is *not* sparse—either because it is genuinely smooth/seasonal or because a very low activity threshold captures many small fluctuations—the eventization step becomes a near-identity transformation. In both cases, BI values track the raw signal and IBG values become nearly constant. Under these conditions, NetBurst’s representation effectively collapses to that of a standard seq2seq forecaster (such as Chronos), and we therefore expected NetBurst to match the performance of existing models rather than introduce new behavior. This implicit hypothesis motivated us to concentrate the evaluation on the sparse, burst-dominated regime, where we expected (and found) the strongest gains. In hindsight, neither explicitly stating nor empirically validating this assumption understandably suggest that NetBurst is specialized and does not generalize to dense or noisy settings.
>
> **(b) *Does eventization fail for dense time series or for bursty-but-noisy series with very low $T_{\text{act}}$?***
> No. In both scenarios, the eventization step essentially passes through the original structure of the series rather than distorting it. For genuinely smooth or seasonal datasets, every point is treated as a meaningful event, so NetBurst receives a representation that is effectively identical to the raw time series. Likewise, for bursty but noisy datasets where $T_{\text{act}}$ is set very low, the BI/IBG decomposition collapses into a dense, high-resolution representation that preserves all fine-grained fluctuations. In both cases, NetBurst naturally reduces to behaving like a conventional continuous forecaster rather than a specialized point-process model.
>
> **(c) *What do the empirical results show?***
> Following your question (and similar concerns from other reviewers), we evaluated NetBurst on the same smooth and seasonal benchmark datasets used by Chronos. As hypothesized, NetBurst performs nearly identically to Chronos on these datasets, confirming that it does not lose predictive power or fail when the series is dense or noisy. These experiments validate the assumption that eventization does not harm performance outside the burst-dominated regime.
>
> **(d) *How will we revise the paper?***
> We will clarify how eventization behaves in dense and noisy regimes, include the new dense-dataset results, and make explicit that NetBurst gracefully falls back to a standard forecaster outside the sparse, bursty setting while delivering real gains where traditional methods underperform.

---

> ### Author Response · Authors · 2025-11-21
> **Response to Q3 and Q5**
>
> **Q3:** *The Oracle analysis in Figure 5b is very insightful. It shows that timing is the dominant error source for sparse Service traces. Why do you think modern Transformer architectures struggle so much with predicting the timing (IBG) of these rare events, and does your work suggest a need for fundamentally different temporal modeling approaches beyond standard self-attention?*
>
> **Q5:** *The dual autoregressive models for IBG and BI are trained independently. Is there any information lost by ignoring the potential statistical dependence between the timing of a burst and its magnitude (e.g., do longer idle periods tend to be followed by larger bursts)? Have you explored coupling the two models, for instance, by conditioning the BI prediction on the predicted IBG?*
>
> ---
>
> ## **Answer A3&A5:**
>
> Thank you for these thoughtful questions. Below we clarify why timing (IBG) is difficult to learn, how this relates to the reviewer’s intuition about dependencies between IBG and BI, and how this feedback from the reviewer motivated us to conduct new experiments that strengthen our results.
>
> **(a) Why do modern Transformers struggle with predicting IBG in sparse, bursty telemetry?**
> IBG sequences in service-level telemetry data are extremely intermittent and skewed: long runs of identical values are punctuated by rare transitions that precede bursts. This structure provides very limited signal for a Transformer trained only on IBG tokens—the majority of observations look identical, and the informative variations are too infrequent to reliably condition on. As a result, the model has difficulty learning when a rare transition is likely to occur, which contributes to timing being the dominant source of error in Figure 5b.
>
> **(b) Does this imply that the timing and magnitude of bursts may be related?**
> Yes, and the reviewer’s comment directly motivated us to articulate and examine this relationship more carefully. In many networked systems, operational dynamics naturally couple idle durations and burst sizes—longer idle periods often precede larger bursts, while shorter idle periods may precede smaller ones. This observation suggests a clear hypothesis: explicitly modeling the dependency between IBG and BI should make IBG prediction more effective, particularly in sparse regimes where IBG alone provides limited signal. Training the two models independently risks discarding these cross-series cues, which may contain exactly the predictive information missing from IBG by itself.
>
> **(c) How did we validate this hypothesis, and what did we observe?**
> Motivated by this question, we explored a **twin-head variant of NetBurst** in which BI and IBG representations are merged and passed through a shared backbone, allowing the model to learn relationships between the decomposed series. Our hypothesis—consistent with the reviewer’s intuition—was that learning these relationships should improve IBG prediction. Our preliminary experiments confirmed this: the twin-head architecture yields further improvements in both MASE and WD compared to the original independent models, demonstrating the value of preserving and learning the inherent dependencies between BI and IBG.
>
> **(d) How will we update the paper?**
> We will add this coupled-model experiment as a separate subsection in the revised version and highlight that learning cross-series relationships meaningfully improves timing accuracy. This extension strengthens the paper’s contribution and directly reflects the insight surfaced by the reviewer’s question.

---

> ### Author Response · Authors · 2025-11-21
> **Response to Q4**
>
> **Question Q4:** *By collapsing each burst into a single BI value, the model discards information about the burst's duration and internal shape. In what operational scenarios would this be a critical limitation, and how could the NETBURST framework be extended to model or generate richer, intra-burst dynamics?*
>
> **Answer A4:**
> Your insightful observation points to an important aspect that we didn’t adequately address in the paper. Below we clarify the scope of limitation you point to, articulate our cross-scale generalizability hypothesis, and explain how our framework can be extended to capture intra-burst dynamics without architectural changes.
>
> **(a) Is collapsing each burst into a single BI value a limitation?**
> Yes—using one BI value *does* discard the burst’s internal shape. However, this limitation is inherent to any temporal aggregation procedure: when simply aggregating 1 sec windows into 1 min or 1 min windows into 1 hour, fine-grained intra-window structure is lost. NetBurst does not merge multiple long bursts into a single event; it simply removes sub-threshold windows so the model focuses on statistically meaningful fluctuations in sparse telemetry data. Whether intra-burst dynamics are preserved depends entirely on the temporal resolution at which NetBurst is applied, not on the architecture itself.
>
> **(b) In what scenarios might this matter?**
> This becomes a limitation when operators require micro-level burst structure—for example, identifying spikes *within* a long surge for anomaly diagnosis or capacity planning. At coarse temporal granularity, that information is unavailable, just as it would be in any aggregated time-series representation.
>
> **(c) Can NetBurst model intra-burst structure if needed?**
> Yes. A key advantage of NetBurst is that it is not tied to a single temporal granularity. *The same architecture can be applied at finer or coarser resolutions without modification.* In our experiments, we demonstrate that NetBurst performs consistently across multiple temporal scales for the same spatial unit, indicating cross-scale generalizability.
>
> **(d) Why does NetBurst generalize across scales?**
> Our hypothesis—aligned with long-standing results on pronounced self-similar scaling characteristics in network traffic—is that telemetry data often exhibits structural regularities across temporal resolutions. Because NetBurst operates using distributional normalization and discrete BI/IBG vocabularies rather than scale-dependent magnitudes, it captures relationships that persist across scales. As a result, intra-burst patterns visible at fine granularity can be learned even when those bursts appear as single BI values at coarser granularity.
>
> **(e) What is the operational implication?**
> This cross-scale behavior supports a practical *reactive zoom-in workflow*: operators can run NetBurst at coarse resolution to flag interesting or anomalous bursts, and then re-run the same model at finer granularity to learn intra-burst dynamics on demand. This enables a principled balance between scalability and interpretability without requiring the development of multiple specialized models.
>
> **(f) How will we revise the paper?**
> We will clarify this multi-resolution capability (currently hidden in Section 4.4) and explain how NetBurst can recover intra-burst structure through finer-scale application rather than architectural changes, addressing the reviewer’s concern directly.

---

> ### Author Response · Authors · 2025-11-21
> **Response to Q6**
>
> **Question Q6:** *Your work draws a powerful connection to Mandelbrot's studies. Beyond network telemetry, what other specific domains (e.g., in finance, climate science, or social networks) do you see as the most promising and immediate applications for the NetBurst framework, and what new challenges might arise in those contexts?*
>
> **Answer A6:**
> Your point is well taken, and we agree that it is important to clarify why packet-level telemetry data was our primary evaluation domain and how NetBurst generalizes to other bursty observational settings. Below, we explain our dataset choices, outline where NetBurst naturally applies, and present new cross-domain results generated in response to your question.
>
> **(a) Why did we focus on packet-level telemetry data?**
> Packet-level telemetry data offers the rare combination needed to train and evaluate a transformer-based forecaster in a burst-dominated regime: (i) large-scale, publicly accessible datasets; (ii) naturally sparse and heavy-tailed behavior aligned with the BI/IBG factorization; and (iii) continuous-valued burst magnitudes suitable for regression. Other observability data (e.g., SNMP counters, system logs) would benefit from NetBurst but are generally not publicly available, making them unsuitable for reproducible evaluation.
>
> **(b) Does NetBurst extend beyond networking?**
> Yes. Sparse, burst-driven time series appear in many fields—wildfire activity, extreme climate events, epidemiological surges, financial volatility spikes, and social behavior dynamics. However, most public datasets in these domains are structured as classification or binary-event problems rather than regression over burst magnitudes, which makes them incompatible with NetBurst’s forecasting formulation. This practical constraint shaped our choice of telemetry data as the primary domain.
>
> **(c) Did we evaluate NetBurst outside networking despite these constraints?**
> Yes. Yours and other reviewers’ comments motivated us to examine NetBurst’s behavior outside networking. To do this systematically, we looked for non-networking datasets that satisfy the three conditions required for NetBurst’s eventization: (1) a sparse and bursty structure, (2) continuous-valued event magnitudes suitable for BI regression, and (3) public availability for reproducible evaluation. Among commonly used observational datasets, wildfire activity is one of the few that meets all three criteria. This made it a principled rather than arbitrary choice for testing BI/IBG beyond packet telemetry. Applying the exact same BI/IBG pipeline and NetBurst architecture, we found that NetBurst outperformed baselines on the wildfire dataset. More concretely, NetBurst achieved a MASE of 0.8145, about 20% lower than the strongest baseline (LagLlama), while maintaining a similar WD of 0.2417. This experiment confirms that both the preprocessing step and the modeling approach transfer naturally to other burst-dominated regimes when the underlying statistical regime matches the conditions NetBurst is designed for.
>
> **(d) What challenges arise in applying NetBurst to other domains?**
> Domains like climate or finance often involve additional exogenous drivers or multivariate features. Applying NetBurst there may require architectural extensions such as cross-attention across features, but the BI/IBG factorization remains directly applicable.
>
> **(e) How will we revise the paper?**
> We will revise the manuscript to (1) clarify why packet-level telemetry data was the natural and practical starting point; (2) discuss the broader applicability of NetBurst and why suitable public regression datasets are limited outside networking; and (3) include the wildfire results to demonstrate concrete cross-domain generalizability.

---

> > ### Comment · Reviewer_DzR8 · 2025-11-26
> >
> > Thank you for your response, here below the summary of what I think the paper is missing.
> >
> > **Re: Question 1 (Sensitivity to `Tact`)**
> > The new sensitivity analysis on `Tact` is a necessary addition. The finding that performance (MASE) improves as the threshold increases is a non-trivial result that should be in the main paper, as it implies that the model benefits from a sparser, more curated set of "events." However, this also raises a new question: what is the limiting behavior? Does this trend continue indefinitely, suggesting that one should always choose the highest possible threshold that still yields enough events for training? The guidance to use a "60–80th percentile" rule is a practical heuristic, but it lacks a formal justification and does not address how the optimal percentile might vary across datasets with different underlying sparsity levels. The sensitivity to this manual choice remains a methodological weakness.
> >
> > **Re: Question 2 (Performance on Dense Benchmarks)**
> > The new experiment showing that NETBURST performs "nearly identically to Chronos" on standard dense benchmarks is a critical piece of evidence. However, this result should be interpreted with caution. It demonstrates that the eventization step does not *harm* performance in this regime, but it also shows that it provides no *benefit*. This confirms that NETBURST is not a general-purpose improvement over existing models; rather, it is a specialized architecture whose advantages are confined to a specific (bursty, intermittent) data regime. While it "gracefully falls back," it does not advance the state-of-the-art on these widely-used benchmarks. This context is essential and must be made explicit to avoid giving the impression of universal superiority.
> >
> > **Re: Questions 3 & 5 (Timing Prediction and Model Coupling)**
> > The hypothesis that Transformers struggle with IBG prediction due to low-signal, highly skewed token distributions is plausible. The new experiment with a "twin-head" variant that couples the IBG and BI streams is a significant finding. The fact that this coupled model yields *further improvements* is a strong result. However, it also serves as a direct criticism of the original, simpler design presented in the main paper. It suggests that the initial choice to model the two streams independently was suboptimal. The paper's core methodology is now in question: is the correct design the dual independent model, or this new, superior coupled model? The paper must be updated to reflect that the independent-stream approach is demonstrably not the best-performing variant of the NETBURST framework.
> >
> > **Re: Question 4 (Intra-Burst Structure)**
> > The argument that intra-burst information loss is a function of temporal aggregation, not the model itself, is a valid reframing. The proposed "reactive zoom-in" workflow is a practical, but reactive, solution. It does not address the fundamental limitation that the model, at any given resolution, is incapable of capturing the *shape* or *evolution* of a burst. It treats all events as instantaneous spikes. For any operational task that depends on the profile of a burst (e.g., distinguishing a sudden shock from a sustained ramp-up), the current model is inadequate. This is a significant architectural limitation that is not resolved by simply changing the input resolution.
> >
> > **Re: Question 6 (Generalization Beyond Networking)**
> > The new experiment on a wildfire dataset is a welcome addition to test cross-domain generalization. A 20% MASE improvement over the strongest baseline is a solid, but not order-of-magnitude, result. While this demonstrates transferability to another bursty domain, the search for a dataset that fits the model's narrow requirements (sparse, continuous-valued, public) highlights the specialized nature of the approach. This finding tempers the broader claims of applicability made in the paper's introduction and conclusion. The evidence supports the claim that NETBURST works well for a specific *statistical regime*, not necessarily for a wide variety of domains.
> >
> > **Conclusion:**
> > The response and new experiments have clarified the scope of the paper's contributions, but they have also highlighted several of its limitations more sharply. The sensitivity to the `Tact` threshold remains a practical concern. The new "twin-head" experiment suggests the core model design is suboptimal. The model's inability to handle intra-burst structure is a key architectural limitation. Finally, the results show that the model's significant advantages are confined to a very specific type of data, and its performance on standard benchmarks is merely on-par with existing methods. The paper presents a strong result for a niche problem, but the claims of broader impact and generality must be significantly toned down.

---

> ### Author Response · Authors · 2025-12-02
>
> We thank the reviewer for the thoughtful follow-up and for articulating the remaining concerns so clearly. We agree that several clarifications must be made more prominent in the revised manuscript, and we will update the paper accordingly.
>
> First, regarding $T_{\text{act}}$, our response may have unintentionally framed this threshold as a model-sensitivity issue rather than what it actually represents in practice: an *operational* choice. $T_{\text{act}}$ governs which fluctuations a practitioner considers meaningful for their domain. Some telemetry series, such as the service-level time series datasets in PINOT and MAWI traces, are intrinsically sparse and bursty; others  (e.g., IP/subnet-level time series datasets) appear dense only because of abundant low-level noise that has little operational value. Raising $T_{\text{act}}$ does not “tune” NetBurst for better performance—it simply restricts the definition of what the practitioner considers an event. The sensitivity analysis demonstrates that NetBurst behaves smoothly across thresholds of practical relevance and that the model does not rely on a fine-tuned choice. We will clarify this framing explicitly, and we will also note that while our percentile-based rule is a simple, reproducible heuristic, more formal or learned approaches are promising future directions.
>
> On dense benchmarks, we agree that the revised version must clearly state that NetBurst is not intended to advance the state of the art for smooth, seasonal series. However, the key point that became lost is that NetBurst is *complementary* to conventional time-series models, not a replacement for them. In dense regimes, the eventization step becomes a near-identity transform, so NetBurst naturally collapses to the behavior of a standard continuous forecaster. The important finding is not that it fails to improve over Chronos, but that it does not degrade performance. This is by design: NetBurst retains the predictive power of established models where they are strong and provides significant gains precisely in the regime where they are not. We will make this complementarity explicit in the revised narrative.
>
> With respect to Questions 3 and 5, the reviewer is correct that the coupled (twin-head) variant yields further gains. Rather than undermining the original design, this result reinforces the modular nature of the NetBurst pipeline. The independent-stream architecture provides a clean conceptual decomposition that isolates timing and magnitude; at the same time, the pipeline intentionally allows the prediction module to be swapped out. The twin-head model is simply a stronger instantiation of that module. We will revise the paper to treat it as an improved variant within the framework rather than as a contradiction of the underlying methodology.
>
> Regarding intra-burst structure, we will clarify the limitation directly. At any fixed temporal resolution, NetBurst treats events atomically, and this is no different from what occurs under any temporal aggregation scheme. The architecture itself does not prohibit intra-burst forecasting; the temporal resolution chosen by the practitioner determines the granularity of the events. Our cross-scale experiments already show that the same model can operate consistently at finer resolutions, and we will emphasize how this enables the reactive “zoom-in” workflow without requiring specialized models.
>
> Finally, we appreciate the reviewer’s concern about generalization beyond networking, but we would like to clarify an important point. The narrow set of public datasets suitable for evaluation reflects *what is publicly available for reproducible research*, not the scope of domains where NetBurst is expected to perform well. Many operational domains—including cloud observability, workflow telemetry, distributed systems, financial microstructure data, and sensor networks—contain exactly the sparse, burst-dominated structure NetBurst is designed for, but do not provide large, openly accessible, continuous-valued burst datasets. Our choice of networking and wildfire data reflects feasibility of evaluation, not a limitation of the model, and we will revise the introduction and conclusion to more carefully frame this distinction.
>
> We believe these clarifications will give the paper a more accurate and measured presentation of NetBurst’s contributions: a modular forecasting framework that (i) recovers conventional performance in dense regimes, (ii) provides substantial gains in sparse and burst-dominated regimes (niche yet critical application domains) where existing models struggle, and (iii) maintains architectural extensibility, as illustrated by the coupled variant.
> However, calling the special regime that NetBurst is targeting a "niche problem" (using this reviewer's terminology) is minimizing its importance in many real-world application domains.

---

### Official Review · Reviewer_JZjE · 2025-10-30

**Soundness:** 3
**Presentation:** 2
**Contribution:** 3
**Rating:** 6
**Confidence:** 3

**Summary:**

The authors highlight the challenges of forecasting network telemetry time series - which are highly bursty and intermittent. They propose an architecture NetBurst, an architecture that uses quantile-based codebooks and dual autoregressors that predict when bursts occur and how large they are, which are then combined into a forecast. Compared to other models, NetBurst reduces MASE significantly on service-level timeseries. The authors present ablation analyses on the model.

**Strengths:**

- I like the fact that the authors delved deep into a specific type of time series and collated datasets specific to this domain of time series. We need more such deep-dive analyses instead of generic time series benchmarks (which are also very useful)
- The authors motivate their work very well. The preliminary analyses that they present in Fig 1 with Fano factors, autocorrelation and local example is interesting.
- The presented model is simple and builds on prior work (Chronos) with targeted modifications. Although it deserves better explanation, to my understanding, the model makes sense and doesn’t have any “fluff”.
-  I like the analyses such as “Do quantile codebooks improve fidelity over uniform binning?”. It shows that the authors attempted to deeply understand the model’s results.

**Weaknesses:**

- Many parts of the paper are unclear, all of which cloud my evaluation of the paper. I mention all these in the questions section. The only reason for my score of 6 is the presentation which is very underwhelming.

**Questions:**

- Are the time series foundation models (TSFMs) (Chronos, Lag-Llama) being finetuned on the given datasets? Does Table 1 present results of these models being finetuned (if it’s a TSFM)? I have the same question for all results presented in the paper. This can be made clear throughout the paper and also in the abstract.

- Suggestion: The authors state “Continuous-time point-process models capture sparsity but fail to handle heavy-tailed magnitudes and long-range dependence.”. References should be given for those readers who don’t know what continuous time point process models

- “our work is an instance of Mandelbrot meets AI.” I”ve not heard of that. Can you give references?

- The authors claim “Models either dilute bursts among zeros or learn shortcuts, such as predicting zero everywhere. This explains the misleadingly low errors of DeepAR on some sparse series—it minimizes loss by ignoring rare events entirely.” - can the authors give empirical evidence for this? I don’t see any reference to their experiments which gave them this conclusion.

-  “Table 1 illustrates this directly: despite pretraining, Chronos collapses on network telemetry data.” Do you pretrain Chronos from scratch, or finetune it?

- “we fix an activity threshold $T_{act}$ and declare a burst whenever consecutive windows exceed this threshold.” Is this threshold determined per dataset? Is this a hyperparameter? Or is it the same for all dataset?
Also this is a limitation of the framework that should be highlighted.

- If I'm right, the authors pre-process the time series into two time-series that have the inter-burst gap and the burst intensity.
Is this adaptable to other datasets? The framework depends on the data being pre-processed so it would be ideal to have the data pre-processing pipeline adaptable to new datasets.

- If the authors are indeed training all the model, I think it is worth adding zero-shot baselines without training to show how much performance you can get without training. This will help the reader understand why training is necessary to show results on this benchmark.

- As a side note, - I’m curious if event prediction baselines be more appropriate for this paper. I'm curious why the authors did or did not consider them.

---

> ### Author Response · Authors · 2025-11-21
>
> We thank the reviewer for providing substantial feedback in the form of insightful comments and specific questions. We are sorry the reviewer found the presentation unclear and underwhelming but hope that in view of our answers below that detail how we will address each of the reviewer’s questions in the revised manuscript, the reviewer can be persuaded to re-evaluate the contributions of our work.
>
> For ease of navigation, we provide our full rebuttal across the following four comments, in order:
> * Response to Q1-Q4
> * Response to Q5-Q6
> * Response to Q7-Q8
> * Response to Q9

---

> ### Author Response · Authors · 2025-11-21
> **Response to Questions Q1-Q4**
>
> **Question Q1:** *Are the time series foundation models (TSFMs) (Chronos, Lag-Llama) being finetuned on the given datasets? Does Table 1 present results of these models being finetuned (if it’s a TSFM)? I have the same question for all results presented in the paper. This can be made clear throughout the paper and also in the abstract.*
>
> **Answer A1:**
> We agree this issue needs to be stated more clearly. In all our experiments, TSFMs such as Chronos and Lag-Llama are fine-tuned on each target dataset rather than evaluated in zero-shot mode. For Table 1 and throughout the paper, the reported TSFM numbers correspond to models **initialized** from their public pretrained checkpoints (when available) and then **fine-tuned** on the dataset at hand. Non-TSFM baselines are trained from scratch using the architectures and training setups described in their original papers. In the revised manuscript, we will explicitly indicate in both the abstract and in Section 4: (1) which models are TSFMs, and (2) that they are fine-tuned on each dataset. Additionally, in Table 1, we will include zero-shot numbers for TSFMs.
>
> ---
>
> **Question Q2:** *Suggestion: The authors state “Continuous-time point-process models capture sparsity but fail to handle heavy-tailed magnitudes and long-range dependence.” References should be given for those readers who don’t know what continuous time point process models are.*
>
> **Answer A2:**
> We will follow this suggestion and will add explicit references and context. In the revised version, we will (i) briefly define continuous-time point process models (e.g., Hawkes processes and their neural extensions) and (ii) cite representative works that motivate their use for sparse event sequences and discuss their limitations when modeling heavy-tailed marks and long-range dependence. We will soften the wording to avoid over-generalization and instead emphasize that **standard marked Hawkes formulations with categorical marks** are poorly matched to our regime of continuous, heavy-tailed burst intensities, and we will support this with both references and our own empirical results that we obtained when also evaluating a new Hawkes process baseline.
>
> ---
>
> **Question Q3:** *“our work is an instance of Mandelbrot meets AI.” I’ve not heard of that. Can you give references?*
>
> **Answer A3:**
> You are correct that "Mandelbrot meets AI" is an informal phrase that we coined in this paper, not a standard term in the literature. Our intention to use this phrase was twofold. First, to acknowledge Mandelbrot’s foundational contributions to the statistical regime that consists of time series with sparse, bursty, and heavy-tailed characteristics, has been encountered in many real-world datasets (including network traffic), and has motivated much of our work, and second to position NetBurst as a way to bring attention to this statistical regime and the challenges it poses for modern sequence models. When revising the paper, we will add concrete references to Mandelbrot’s work and where it starts to intersect with learning-based efforts and will make sure the text does not suggest that "Mandelbrot meets AI" is an established or common term.
>
> ---
>
> **Question Q4:** *The authors claim “Models either dilute bursts among zeros or learn shortcuts, such as predicting zero everywhere. This explains the misleadingly low errors of DeepAR on some sparse series—it minimizes loss by ignoring rare events entirely.” – can the authors give empirical evidence for this? I don’t see any reference to their experiments which gave them this conclusion.*
>
> **Answer A4:**
> Thank you for catching this. We acknowledge that our wording here was too strong and not sufficiently tied to explicit evidence. Our observation is that on extremely sparse telemetry data, **sequence-level MASE over all timesteps** is dominated by long runs of zeros. In this regime, we empirically see DeepAR (and similar baselines) produce forecasts that stay near zero, achieving deceptively low MASE while performing poorly on the rare bursts themselves. This behavior is what we meant by “diluting bursts among zeros”. To avoid any misunderstanding, we will (i) soften the language to describe this as an **empirical behavior we observe in our experiments**, rather than an universal “shortcut”, and (ii) add a pointer to specific results (e.g., a figure or table comparing event-only MASE vs full-series MASE for DeepAR and NetBurst) to substantiate the claim. If space permits, we will also include a small diagnostic plot showing DeepAR’s predicted vs true values on a sparse time series to illustrate this effect.

---

> ### Author Response · Authors · 2025-11-21
> **Response to Questions Q5-Q6**
>
> **Question Q5:** *“Table 1 illustrates this directly: despite pretraining, Chronos collapses on network telemetry data.” Do you pretrain Chronos from scratch, or finetune it?*
>
> **Answer A5:**
> We do **not** pretrain Chronos from scratch. We use the **public Chronos checkpoints provided by the authors and fine-tune them** on our telemetry datasets following their recommended training setup. Our point was that even with pretraining on broad time series corpora, Chronos performs poorly on our sparse, burst-dominated telemetry regime. We will rephrase this sentence to avoid any ambiguity—for example: “Even when initialized from its pretrained checkpoint and fine-tuned on our telemetry data, Chronos underperforms substantially in the sparse, bursty regime we study.” We will also ensure that all mentions of Chronos and Lag-Llama clearly state “pretrained + fine-tuned” rather than “pretrained” alone.
>
> ---
>
> **Question Q6:** *“we fix an activity threshold and declare a burst whenever consecutive windows exceed this threshold.” Is this threshold determined per dataset? Is this a hyperparameter? Or is it the same for all dataset? Also this is a limitation of the framework that should be highlighted.*
>
> **Answer A6:**
> Thank you for raising these clarifying questions. We address each of them below.
>
> **(a) Is $T_{\text{act}}$ determined per dataset?**
> Yes. $T_{\text{act}}$ is selected separately for each dataset. Different telemetry time series vary widely in scale, sparsity, and noise characteristics, so a single global threshold would suppress meaningful bursts in some datasets and elevate noise in others. A per-dataset choice is therefore necessary for meaningful eventization.
>
> **(b) Is $T_{\text{act}}$ a hyperparameter?**
> Yes. $T_{\text{act}}$ is a data-driven hyperparameter. In practice, we use a simple, automatic procedure: we compute $T_{\text{act}}$ from a short calibration window by selecting a value between the 60th and 80th percentile of non-zero activity. This approach keeps the thresholding process scale-aware without requiring manual, domain-specific tuning.
>
> **(c) Is $T_{\text{act}}$ the same across all datasets?**
> No. Because the underlying activity distributions differ significantly across datasets, $T_{\text{act}}$ is computed separately for each dataset using the same percentile-based rule. This ensures that “activity” is consistently defined relative to each dataset’s statistical structure.
>
> **(d) Is the need for $T_{\text{act}}$ a limitation of the framework?**
> We view $T_{\text{act}}$ not as a fundamental limitation but as a deliberate feature of the framework. Different operational settings care about different event magnitudes, and giving practitioners control over what constitutes a meaningful event is often desirable. Determining the operationally relevant activity range is a standard part of exploratory data analysis for any new time series, and our automatic percentile-based rule provides a simple, reproducible default. While $T_{\text{act}}$ is an extra parameter, its interpretability and adaptability make it aligned with practical workflows rather than a restrictive limitation.
>
> **(e) How sensitive is NetBurst to this choice?**
> Our underlying hypothesis was that NetBurst should behave predictably across a broad range of $T_{\text{act}}$ choices: increasing $T_{\text{act}}$ should make the series sparser and improve NetBurst’s absolute performance by filtering noise, and the model’s relative advantage over baselines should persist or grow in higher-sparsity regimes. To evaluate this, we performed a sweep of $T_{\text{act}}$ across a wide range of percentiles and measured performance at each setting. The results confirmed our hypothesis: NetBurst’s performance evolved smoothly with $T_{\text{act}}$, its absolute accuracy improved at higher thresholds, and it consistently outperformed baselines across the sweep. This demonstrates that NetBurst is robust to variations in $T_{\text{act}}$ of practical interest and that practitioners can adjust the threshold to match their operational notion of meaningful events without destabilizing the model.
>
> **(f) How will we revise the paper?**
> We will (i) clearly document how $T_{\text{act}}$ is chosen in practice, (ii) emphasize that it is a per-dataset, data-driven hyperparameter, (iii) present the full sensitivity analysis, and (iv) explain how the threshold provides a flexible way for practitioners to define operationally meaningful events. We will also note that learned or adaptive thresholds are a valuable future extension for event learning.

---

> ### Author Response · Authors · 2025-11-21
> **Response to Questions Q7-Q8**
>
> **Question Q7:** *If I'm right, the authors pre-process the time series into two time-series that have the inter-burst gap and the burst intensity. Is this adaptable to other datasets? The framework depends on the data being pre-processed so it would be ideal to have the data pre-processing pipeline adaptable to new datasets.*
>
> **Answer A7:**
> Yes, it is adaptable. The BI/IBG process is a general eventization template that applies to any univariate time series with a meaningful notion of “activity.” Given a time series and an activity signal (e.g., throughput above a threshold, counts above background noise, or intensity above a physical threshold), bursts can be defined as contiguous segments exceeding that activity level. BI then captures the burst magnitude and IBG captures the time between bursts. The only dataset-specific choices are selecting the activity signal and choosing an appropriate $T_{\text{act}}$; the NetBurst architecture does not need modification. The following is a more granular answer to your question:
>
> **(a) Why did we choose packet-level telemetry as the primary domain?**
> Packet telemetry is one of the few domains where large-scale, publicly available datasets have the exact statistical properties NetBurst is designed for: strong sparsity, large burst intensity variation, and heavy-tailed behavior. It also supports regression over continuous burst magnitudes, which is central to evaluating BI prediction quality. Many other bursty real-world systems exist, but the publicly available datasets in those domains are usually structured as classification or detection tasks rather than regression over event magnitudes. Packet telemetry therefore provides the most appropriate and reproducible starting point.
>
> **(b) How did we demonstrate cross-domain generalizability?**
> Your comment directly motivated us to examine NetBurst’s behavior outside networking. To do this systematically, we looked for non-networking datasets that satisfy the same three conditions required for NetBurst’s eventization: (1) a sparse and bursty structure, (2) continuous-valued event magnitudes suitable for BI regression, and (3) public availability for reproducible evaluation. Among commonly used observational datasets, wildfire activity is one of the few that meets all three criteria. This made it a principled rather than arbitrary choice for testing BI/IBG beyond packet telemetry. Applying the exact same BI/IBG pipeline and NetBurst architecture, we found that NetBurst outperformed baselines on the wildfire dataset. More concretely, NetBurst achieved a MASE of 0.8145, about 20% lower than the strongest baseline (LagLlama), while maintaining a similar WD of 0.2417. This experiment confirms that both the preprocessing step and the modeling approach transfer naturally to other burst-dominated regimes when the underlying statistical regime matches the conditions NetBurst is designed for.
>
> **(c) How will we revise the paper?**
> We will clarify the general applicability of the BI/IBG pipeline, explain why packet telemetry was the natural primary domain, explicitly motivate the choice of the wildfire dataset as a non-networking case that meets the required statistical criteria, and report the results we obtained when using NetBurst on this non-networking dataset to show concrete cross-domain generalizability.
>
> **Question Q8:** *If the authors are indeed training all the model, I think it is worth adding zero-shot baselines without training to show how much performance you can get without training. This will help the reader understand why training is necessary to show results on this benchmark.*
>
> **Answer A8:**
> This is an excellent suggestion. Our primary goal in this paper was to compare NetBurst against strong, trained baselines under matched training conditions, and so we focused on fine-tuned TSFMs and trained-from-scratch non-TSFM models. We agree that including zero-shot results for pretrained TSFMs (where available) would help quantify how much of NetBurst’s gains are due to architecture vs training on the target regime. In the revision, we plan to add zero-shot Chronos and Lag-Llama results (initialized from publicly available checkpoints and evaluated without any fine-tuning) on at least one of our main telemetry datasets. We will clearly label these as zero-shot and contrast them with fine-tuned performance and with NetBurst, to make the added value of training and of the BI/IBG factorization more transparent.

---

> ### Author Response · Authors · 2025-11-21
> **Response to Q9**
>
> **Question Q9:** *As a side note, – I'm curious if event prediction baselines be more appropriate for this paper. I'm curious why the authors did or did not consider them.*
>
> **Answer A9:**
> Thank you for raising this important question. We agree that event-centric models are natural baselines for a framework that converts a time series into events. To make this comparison explicit, we implemented a strong Hawkes-type baseline and evaluated it directly on our datasets. The following provides a more detailed description of this effort.
>
> **(a) What Hawkes model did we evaluate?**
> We used the transformer-based Hawkes process model proposed in prior work, which represents one of the strongest neural Hawkes architectures available. Because this model is designed for categorical marks (e.g., “small” vs. “large” events), we adapted it to our setting by replacing its classification head with a regression head so that it could predict continuous burst intensities, which are required for BI forecasting.
>
> **(b) What did the empirical results show?**
> Our experiments confirmed our unstated hypothesis. The Hawkes baseline performed very poorly on both datasets we evaluated (MASE > 19.48, WD > 0.3176), failing to capture the magnitude of large bursts and producing unstable predictions. In contrast, NetBurst’s quantile-based preprocessing converts continuous burst magnitudes into discrete bins that are more amenable to sequence modeling and dramatically stabilizes learning.
>
> **(c) Why did we not include more event-centric baselines?**
> Standard continuous-time point-process models assume discrete event types and typically work best when event marks fall into a small set of categories. In contrast, our telemetry bursts have continuous magnitudes with heavy-tailed distributions (very large Fano factors). When a Hawkes model must directly regress these highly variable burst intensities, the mark distribution becomes difficult to model and the prediction task becomes unstable. This mismatch formed the basis of our initial hypothesis that Hawkes models would underperform in our setting, and we internalized that a full empirical comparison might not be necessary.
>
> **(d) How will we revise the paper?**
> We will add the Hawkes baseline to the main text, clearly label it as an event prediction model, describe how we adapted it to continuous marks, and report its performance on one of our telemetry time series. This will make the comparison explicit and transparent and will help clarify why NetBurst’s preprocessing strategy is better suited to the statistical regime encountered for telemetry data.

---

> > ### Comment · Reviewer_JZjE · 2025-11-26
> >
> > Thank you for the responses.
> >
> > I'm convinced with the authors' answers and maintain my score.

---

### Meta-Review · Area_Chair_WRD7 · 2026-01-04

**Summary:**

The reviewers generally recognized the proposed NetBurst framework for its specialized focus on the challenging regime of bursty, intermittent network telemetry data. Key strengths highlighted include the fundamental rethinking of forecasting as an event-centric paradigm and the improved performances.

However, several concerns exist: reviewers pointed to bad presentation and formatting issues. Also, there were significant worries regarding the narrow applicability of the method. In summary, it is hard to justify the true significance of the contribution in the machine learning community.

**Reviewer Concerns:**

The authors implemented a Transformer Hawkes Process with a regression head, demonstrating that it underperforms significantly in this heavy-tailed regime. They included evaluations on dense benchmarks (showing NetBurst matches Chronos) and a wildfire dataset.  A sensitivity analysis was provided.

There are some outstanding concerns. While the authors argue that this is a "resolution choice," the critique that the model collapses complex bursts into single values may remain a fundamental architectural limitation, as noted by Reviewer DzR8.

Despite the rebuttal's clarity, the original manuscript’s formatting were mentioned as not good by several reviewers.

The core innovation remains largely a preprocessing step. Reviewers thought that this may be more suited for a specialized networking/systems venue rather than the general ICLR conference.

**Reviewer Scores:**

Reviewers may keep their original scores.

---

### Decision · Program_Chairs · 2026-01-26

Reject